# Atomistic insights into highly active reconstructed edges of monolayer 2H-WSe$_2$ photocatalyst

Mohammad Qorbani[1,2], Amr Sabbah [3,4,12], Ying-Ren Lai[1,2,12], Septia Kholimatussadiah[1,5,6,7], Shaham Quadir[1,3,8,9], Chih-Yang Huang[1,3,8,10], Indrajit Shown[3,11], Yi-Fan Huang[3], Michitoshi Hayashi [1,2], Kuei-Hsien Chen [1,3,13] & Li-Chyong Chen [1,2,5,13]

Ascertaining the function of in-plane intrinsic defects and edge atoms is necessary for developing efficient low-dimensional photocatalysts. We report the wireless photocatalytic CO$_2$ reduction to CH$_4$ over reconstructed edge atoms of monolayer 2H-WSe$_2$ artificial leaves. Our first-principles calculations demonstrate that reconstructed and imperfect edge configurations enable CO$_2$ binding to form linear and bent molecules. Experimental results show that the solar-to-fuel quantum efficiency is a reciprocal function of the flake size. It also indicates that the consumed electron rate per edge atom is two orders of magnitude larger than the in-plane intrinsic defects. Further, nanoscale redox mapping at the monolayer WSe$_2$–liquid interface confirms that the edge is the most preferred region for charge transfer. Our results pave the way for designing a new class of monolayer transition metal dichalcogenides with reconstructed edges as a non-precious co-catalyst for wired or wireless hydrogen evolution or CO$_2$ reduction reactions.

[1] Center for Condensed Matter Sciences, National Taiwan University, Taipei 10617, Taiwan. [2] Center of Atomic Initiative for New Materials, National Taiwan University, Taipei 10617, Taiwan. [3] Institute of Atomic and Molecular Sciences, Academia Sinica, Taipei 10617, Taiwan. [4] On leave from Tabbin Institute for Metallurgical Studies, Tabbin, Helwan 109, Cairo 11421, Egypt. [5] Department of Physics, National Taiwan University, Taipei 10617, Taiwan. [6] Nano Science and Technology, Taiwan International Graduate Program, Academia Sinica, Taipei 11529, Taiwan. [7] Institute of Physics, Academia Sinica, Taipei 11529, Taiwan. [8] Molecular Science and Technology Program, Taiwan International Graduate Program, Academia Sinica, Taipei 11529, Taiwan. [9] Department of Physics, National Central University, Taoyuan City 32001, Taiwan. [10] International Graduate Program of Molecular Science and Technology, National Taiwan University, Taipei 10617, Taiwan. [11] Department of Chemistry, Hindustan Institute of Technology and Science, Chennai 603103, India. [12] These authors contributed equally: Amr Sabbah, Ying-Ren Lai. [13] These authors jointly supervised this work: Kuei-Hsien Chen, Li-Chyong Chen. ✉email: chenkh@pub.iams.sinica.edu.tw; chenlc@ntu.edu.tw

Climate change has necessitated the framing of government regulations and the development of green strategies for reducing $CO_2$ emissions. Scientists worldwide are engaged in efforts to find sustainable solutions to the problem of $CO_2$ level in the air[1–3]. Gas-phase photocatalytic $CO_2$ reduction reaction (PC $CO_2$RR) over artificial leaves—artificial leaves fulfills the solar-to-fuel processes of natural photosynthesis—to value-added hydrocarbons is a potential solution[4]. $CO_2$RR through artificial photosynthesis involves a series of processes: light-harvesting, exciton dissociation, photogenerated charge carrier diffusion, and transfer to the adsorbed molecule[4–7]. Researchers have rationally designed different material-based strategies to realize high performance in each step: e.g., introducing co-catalyst[8,9], modifying the electronic structure[10,11], heterostructuring[12–14], and enhancing the surface area[15,16] (Supplementary Fig. 1 and Supplementary Table 1). The strategy of using a co-catalyst (like precious Pt) results in a conversion rate that is higher than that of other strategies by one to two orders of magnitude. In contrast, the lack of activation sites in co-catalyst-free systems results in a low areal conversion rate and efficiency.

Atomically thin transition metal dichalcogenides (TMDCs)—$MX_2$, where M and X are transition metal and chalcogen, respectively—have emerged as a new class of non-precious (photo)catalysts with high activity and strong light–matter interaction[17–20]. It has been visualized that their edge sites[21] and the basal plane of TMDCs (modified by introducing dopants[22] or vacancies[23], and by applying strain[24]) can act as efficient active sites for the hydrogen evolution reaction (HER) and $CO_2$RR. Further, commonly observed intrinsic in-plane point defects—monochalcogen vacancy ($V_X$), dichalcogen vacancy ($V_{X2}$), vacancy complex of metal with three neighboring chalcogens ($V_{MX3}$), vacancy complex of metal with three neighboring chalcogen pairs ($V_{MX6}$), and antisite defects with metal substituting for $X_2$ ($M_{X2}$) or $X_2$ column substituting neighboring metal ($X_{2M}$)[25–27]—can also play a role in $CO_2$ molecule activation. For example, sulfur vacancies in $MoS_2$ nanosheets are shown to be the active sites for $CO_2$ hydrogenation to methanol[28]. Therefore, catalytic competition between the intrinsic in-plane structural defects and the edge atoms still needs to be investigated systematically.

In this study, we report monolayer (ML) $WSe_2$ flakes, as a robust photocatalyst[29], with different micro-scale lateral sizes that were grown uniformly by a low-pressure vapor deposition method. We identified the intrinsic defects in the basal plane, reconstructed configuration, and several structural defects at the edge. The results of density functional theory (DFT) calculations indicated that both regular and imperfect edge configurations can adsorb $CO_2$ more efficiently than the in-plane point defects. PC experiment results show that the $CO_2$-to-$CH_4$ internal quantum efficiency (IQE) is a reciprocal function of the perimeter of the ML flakes, which exhibit a high consumed electron rate per edge atom, notably surpassing those of reported TMDC catalysts. In this paper, we investigate the ensemble average PC behavior of the edge atoms in atomically thin TMDCs.

## Results

### Characterizations of monolayer $WSe_2$ artificial leaf.
Figure 1a, b shows the sizes and uniform distribution of the ML $WSe_2$ flakes (Supplementary Fig. 2). The average perimeter ($\bar{P}$) of the flakes is in the range of 1.5–25 µm. X-ray photoelectron spectroscopy (XPS) analysis was used to identify the oxidation state and Se-to-W ratio. Figure 1c shows the XPS spectra of W $4f$ and Se $3d$ elements with peak positions at 32.3, 34.5, 54.7, and 55.6 eV for W $4f_{7/2}$, W $4f_{5/2}$, Se $3d_{5/2}$, Se $3d_{3/2}$, respectively, representing $W^{4+}$ and $Se^{2-}$ signals[30]. Quantitative compositional analysis reveals

that the flakes are slightly Se-rich ($\delta \leq 0$ in $W_{1+\delta}Se_2$) due to the higher vapor pressure of Se during the growth process (Fig. 1d). The dependence of $\delta$ to $\bar{P}$ was assigned to the intrinsic in-plane defects and Se-terminated edge atoms. Notably, the perimeter-to-area ratio, which is proportional to $\bar{P}^{-1}$, increases at a faster rate than does $\delta$ with decreasing $\bar{P}$, implying that the edge is not fully terminated with Se atoms. Moreover, the triangular shape of the flakes is another indication of the non-stoichiometric growth process[31]. Raman spectra illustrate several vibrational modes as the signature of ML 2H-$WSe_2$ without heating the samples by a laser beam (Supplementary Figs. 3–5)[32]. The absorption spectrum of the ML flakes contains four peaks: A/B and C/D direct excitonic transitions at the $K$ and $M$ points of the Brillouin zone, respectively (Fig. 1e)[20]. Further, it is calculated that the ML $WSe_2$ can absorb ~4.4 % of the irradiated light.

Room-temperature photoluminescence (PL) spectra show a non-symmetric curve with an optical bandgap of ~1.68 eV (Supplementary Fig. 6). The non-symmetric curves are fitted with two Gaussian peaks corresponding to the neutral exciton ($X^0$) and charged exciton (or trion; $X^-$) quasiparticles[33], as well as binding energy $\triangle E_{X^0,X^-} = 29 \pm 2$ meV matched with the $A'_1$ optical phonon[34]. Laser power-dependent PL spectra were also recorded to study the dynamics of charge-carrier recombination (Supplementary Fig. 7). At high laser power, electron transfer from the $K$ to $\Lambda$ points of the Brillouin zone results in quenching of the PL emission by non-radiative recombination processes (Supplementary Fig. 8). Further, the results of the time-resolved PL (TRPL) experiments indicate a prolonged lifetime for the photogenerated electrons and holes of the smaller flakes (Supplementary Fig. 9). This also can be resulted in higher photogenerated carrier density at the edge regions that could increase the possibility of the defective edge sites acting as recombination centers or enhance the probability of charge transfer to the adsorbed $CO_2$. Figure 1f shows a broad peak (D) at around $1.622 \pm 0.002$ eV with $\triangle E_{X^0,D} \approx 100$ meV and it can be assigned to the convolution of defect-related excitons[35]. The relative intensity of peak D increases with decreasing size of the ML flake, which is in agreement with the TRPL curves, i.e. defect states can increase the exciton lifetime. The presence of vacancies and defects were further proved by a comparison between the ML $WSe_2$ grown by vapor deposition and prepared by mechanical exfoliation (Supplementary Fig. 10). Figure 1g and Supplementary Fig. 11 show that peaks D, $X^0$, and $X^-$ blue-shift with decreasing temperature due to the electron–phonon coupling[36].

### Atomic configurations of the edge.
Using high-angle annular dark-field imaging in scanning transmission electron microscopy (HAADF-STEM), we studied the atomic configuration of the ML 2H-$WSe_2$ (i.e. hexagonal polytype phase) flakes transferred onto a transmission electron microscopy (TEM) grid (Supplementary Fig. 12a–c). The selected-area electron diffraction (SAED) pattern shows that the ML flake is highly crystalline and contains different families of spots, $\vec{K}_a$ and $\vec{K}_b = -\vec{K}_a$ (Supplementary Fig. 12d–f)[37]. Supplementary Fig. 13 shows several intrinsic point defects[25–27], including $V_{Se}$ (commonly observed), $V_{2Se}$, $W_{WSeX}$, $W_{Se}$, and $Se_W$, that were non-uniformly distributed in the basal plane of ML flakes with a density of ~$5 \times 10^{13}$ $cm^{-2}$ (i.e., the defect concentration is ~1.5 at%). The results of first-principles DFT calculations show the relaxed defect structures, formation energies ($\triangle \varepsilon_f$s), and density of states (DOSs). Each defect more and less can distort hexagonal lattice, but just $W_{Se}$ defect breaks the three-fold symmetry of $WSe_2$ with W locating closer to two of the three nearest-neighbor W atoms in agreement with other observations[25] (Supplementary Fig. 14). Based on calculated

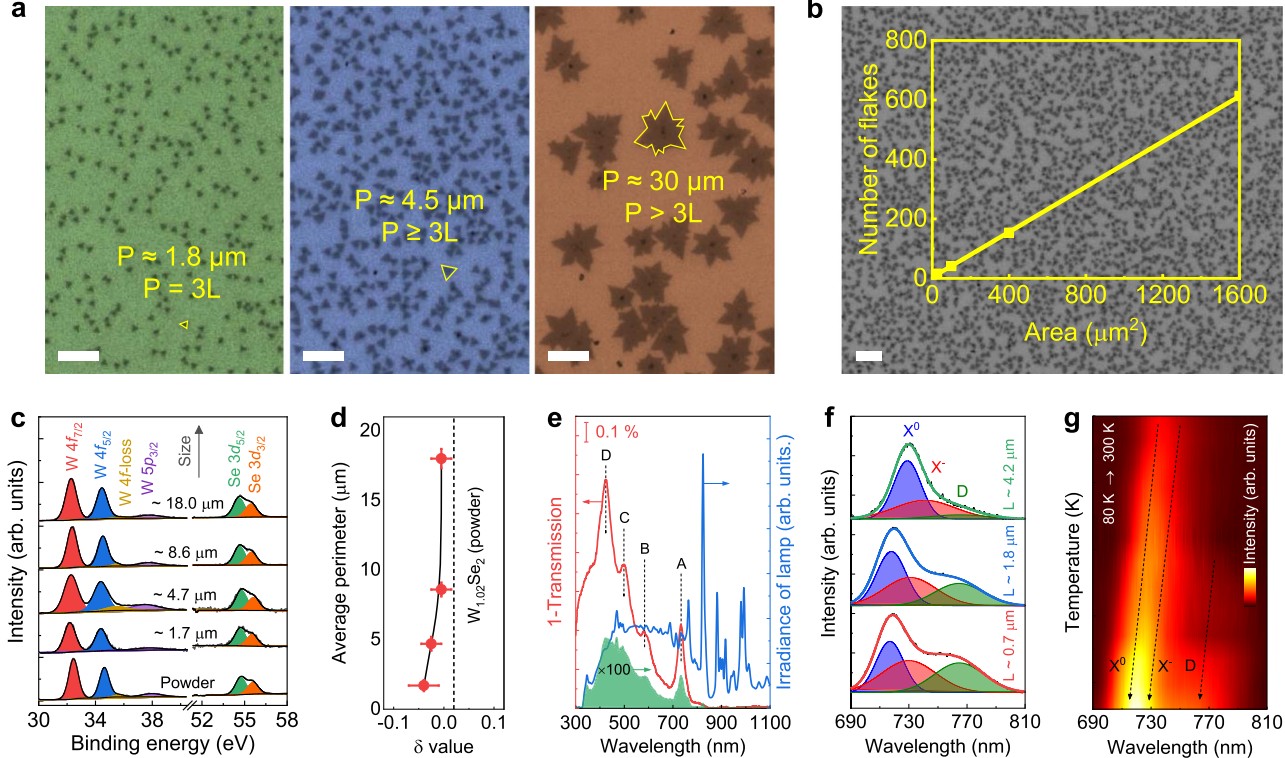

**Fig. 1 Characterization of ML WSe₂ flakes with different sizes. a** Optical microscopy images of flakes with different perimeters (*P*) and characteristic lateral sizes (*L*). Scale bar = 5 μm. **b** Large-area optical microscopy image. Inset shows the linear curve of the number of flakes vs. the area. Scale bar = 5 μm. **c** XPS spectra of W 4f and Se 3d. **d** δ value versus the average perimeter. The line is added as a guide to the eye. **e** Absorption (red line), irradiance of the Xe lamp (blue line) spectrum, and absorbable photon flux (green filled curve). **f** Low-temperature PL spectra of flakes with different lateral sizes fitted with three Gaussian curves. **g** Temperature-dependent PL spectra of a flake with $L \approx 0.7$ μm.

formation energy (Supplementary Fig. 15), $V_{Se}$ is the most probable defect in the entire range of chemical potential of selenium, which is in agreement with the STEM investigation. Supplementary Fig. 16 also shows the formation of defect bands in the gap of the ML 2H-WSe₂.

Figure 2a shows that the edge is not straight at the atomic scale and contains many steps distinguished by different edge angles. Since the edge contains zigzag (ZZ), antenna (An), and armchair (AC) configurations, the actual perimeter of a flake is inherently larger than that measured by optical microscopy images. Theoretically, the average linear densities of edge atoms are about 10 nm⁻¹ (9.1 and 10.6 nm⁻¹ for ZZ and AC edges, respectively). Therefore, the minimum density of the edge atoms is $\sim \frac{2 \times 10^{13}}{P}$ cm⁻² for triangular flakes, where $P$ is in μm. This shows that the densities of the intrinsic defects and the number of edge atoms are in the same range for small flakes. Figure 2b displays the compressive strain up to a maximum of ~14% due to structural reconstructions vertical to the edge due to the imperfect crystal structure[38], which agrees well with the relaxed structure as per our calculations (Supplementary Fig. 17). Figure 2c and Supplementary Fig. 18 also exhibit the calculated edge configurations with defects, which correspond to some of the features seen in STEM images. DFT calculation results revealed that structural defects such as $V_{Se}$ and $W_{Se}$—here, we calculated the point defects at the inert Se-terminated edges—have a lower $\triangle \varepsilon_f$ than in-plane defects, implying that the former can easily form on edges (Fig. 2d). W adatom ($W_{add}$) shows that it is stable at the edge region as compared with the above-mentioned defects. Therefore, in addition to the non-equilibrium condition at the end of the growth process, low $\triangle \varepsilon_f$ may be the reason for the formation of the defects and steps at the edge. Supplementary

Fig. 19 also shows the presence of metallic edge states and the non-zero magnetic moment for the reconstructed ZZ and An edges (commonly observed configurations at the edge)[25,38,39]. However, the reconstructed AC edge indicates a semiconductor nature with a smaller band gap (as compared with the basal plane) and quenched magnetic moments. This suggests that the edge and basal plane can form a local homostructure, which improves charge separation.

**CO₂ adsorption by density functional theory calculation.** We performed DFT calculations to understand the activation mechanism of CO₂ adsorption. The possible active sites are the intrinsic defects on the basal plane, regular edges, and structural defects at the edge (e.g., steps, adatoms, and defects). Supplementary Fig. 20 shows that the basal plane and all the observable intrinsic defects, except $W_{Se}$, indicate physisorption of $CO_2$ molecules with binding energy ($E_b^{CO_2}$) and equilibrium distance ($r_0$, which is the nearest distance between the WSe₂ and CO₂ molecule) of around −0.2 eV and 3.5 Å, respectively. The $W_{Se}$ enables CO₂ binding to form a linear molecule with $E_b^{CO_2} = -0.43$ eV and charge difference $\delta q = 0.03$ $e^-$. Hence, the basal plane of the ML TMDCs is inert, as previously reported[24,40,41], whereas the regular edges are possible active sites for CO₂ adsorption. Figure 3a and Supplementary Fig. 21 show that $ZZ_{Se}$ and $An_{Se}$ edges are also inert with $E_b^{CO_2} \approx -0.2$ eV and $r_0 \approx 3.5$ Å, which are similar to the values for the basal plane. Moreover, reconstructed $ZZ_W$ (both I and II configurations) cannot chemisorb CO₂ because of screening of active anions from the CO₂ by the nearest rearranged Se atoms. In contrast, $An_W$, known as W Klein edge, can chemisorb bent CO₂ depending on

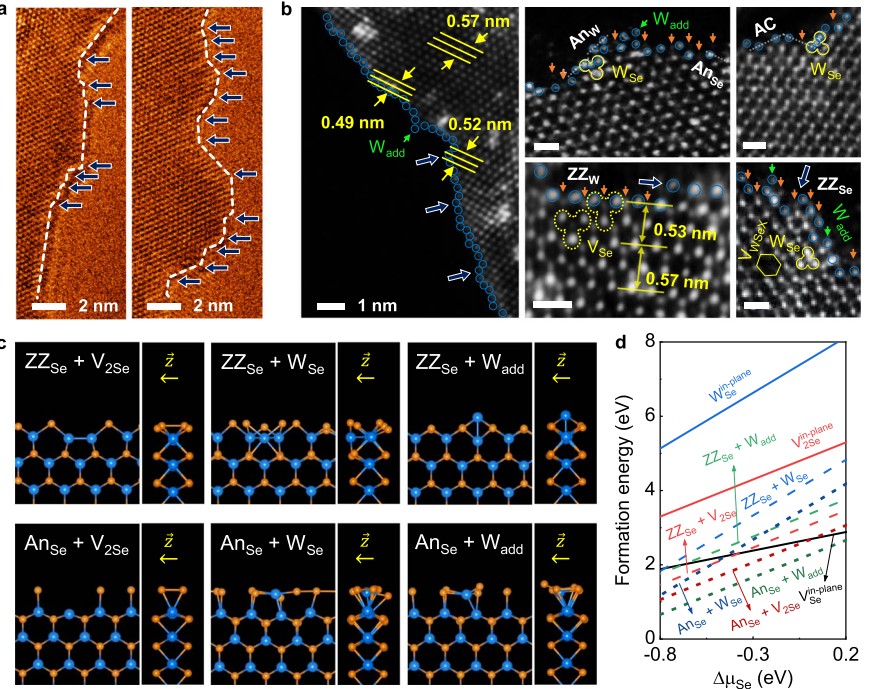

**Fig. 2 Microstructure of basal planes and edges of ML WSe₂ flakes. a** High-resolution TEM images of the typical edges of the ML flakes. **b** HAADF-STEM images of the edge. Scale bar = 0.5 nm. Blue circles (stand for W atoms) are added as a guide to the eye. Dark orange, green, and blue arrows show the dimmer spots for Se, steps, and $W_{add}$ defects, respectively. Hexagons, yellow dashed, and solid three petal flowers show $V_{WSeX}$, $V_{Se}$, and $W_{Se}$ defects, respectively, respectively. **c** Top-view and cross-sectional images of the relaxed structures of the Se-terminated ZZ and An edges with $V_{2Se}$, $W_{Se}$, and $W_{add}$ defects. Blue- and dark orange-filled circles stand for W and Se atoms. **d** Formation energies of Se-terminated edge defects as functions of chemical potential of Se ($\triangle\mu_{Se}$).

the initial conditions (I and II configurations), resulting in higher catalytic activity[42]. Similarly, W atoms at the AC edge can bind an oxygen atom to form linearly chemisorb $CO_2$ with $E_b^{CO_2} = -0.43$ eV. In addition to the regular configurations, structural defects at the edge provide a strong tendency toward $CO_2$ adsorption. For example, $ZZ_{Se}$ (with $W_{Se}$ or $W_{add}$) and $An_{Se}$ (with $V_{2Se}$, $W_{Se}$, or $W_{add}$) can effectively bind $CO_2$ (Fig. 3b and Supplementary Fig. 22). This implies that atomic-scale steps, adatoms, and defects are highly active sites for photocatalytic $CO_2$RR. However, many different kinds of possible structural defects (e.g. $V_W$, $V_{WSeX}$, $W_{Se}$, $V_{Se+Se}$, etc.) exist at the W- and Se-terminated edges; these will be investigated in our future studies.

**Nanoscale redox mapping and photocatalytic performance.** We performed photodeposition experiments for the initial observation of the charge transfer capability of the ML WSe₂. Figure 4a, b show the field-emission scanning electron microscope (FESEM) images of the control sample (in dark) and after Ag photodeposition (under light). These experiments illustrate that the rate of the photodeposition of the Ag nanoparticles is higher at the edge. Besides, Fig. 4c–e shows the atomic force microscopy (AFM) height profile, AFM-based scanning electrochemical microscopy (SECM) feedback maps for main and lift scans of the ML WSe₂, respectively. The positive feedback current ($I_f$) directly illustrates the local electrochemical activity (charge transfer and charge transport) of the sample that is correlated to the charge diffusion in the ML flakes and charge transfer from WSe₂ to the electrolyte (i.e., WSe₂–liquid interface). The $I_f^{main}$ gradually increases from the background fluctuations ($I_{f,SiO_2}$ of the tip on the SiO₂ substrate) to a maximum of 4.2 pA at ~0.2 μm inside the flake (Supplementary Fig. 23). $I_f^{main}$ reaches 2.9 pA at the basal

plane far from the edge of the flake, indicating that the interior region of the ML WSe₂ has little or no activity[43]. As shown in Supplementary Fig. 24, the SECM normalized response versus the tip-surface distance was also fitted with an analytical approximation[44,45]. Results reveal that the edge behaves like a metal albeit with a less charge transfer rate assigned to the spatial confinement in the across direction which is entirely different from the response of the basal plane. These results suggest that the superior electronic properties of the edge atoms can provide efficient charge transfer and transport, resulting in higher catalytic activity.

Gas-phase PC experiments were carried out on the ML WSe₂ photocatalyst without transferring them onto any conductive substrate. Results show the presence of CH₄ and CH₃CHO as the major (selectivity ≥ 90%) and minor products, respectively (Supplementary Fig. 25). Figure 4f shows that the blank-corrected total CH₄ yield ($Y$) depends on the total area ($S_t$) and $\bar{P}$ of ML WSe₂ flakes. Referring to the recently published papers[9,11,28,46] and the absence of CO product (Supplementary Fig. 26), $CO_2$ reduction could be explained by the following reduction reaction pathway: $CO_2$ + * (forming $CO_2$* where the asterisks denote catalytically active sites) followed by generation of COOH*, CO* + H₂O, CHO*, CH₂O*, CH₃O*, and CH₄↑ + O*, OH*, H₂O + * due to the subsequent proton and electron transfer where and the upward vertical arrow represent the release of gas in the reaction. Accordingly, O₂ is the oxidation product due to the hole and H₂O reaction (Supplementary Fig. 27)[11,47]. Notably, $Y$ is proportional to $S_t$ for a certain $\bar{P}$, while a monotonic increase in $Y$ with decreasing $\bar{P}$ is observed. Based on these observations, we suggest an ensemble average model for the total yield as a sum of the contributions of the basal plane and the edge: $Y = (f_b + \frac{f_e}{\bar{P}})S_t$, where $f_b$ and $f_e$ are fitting factors corresponding

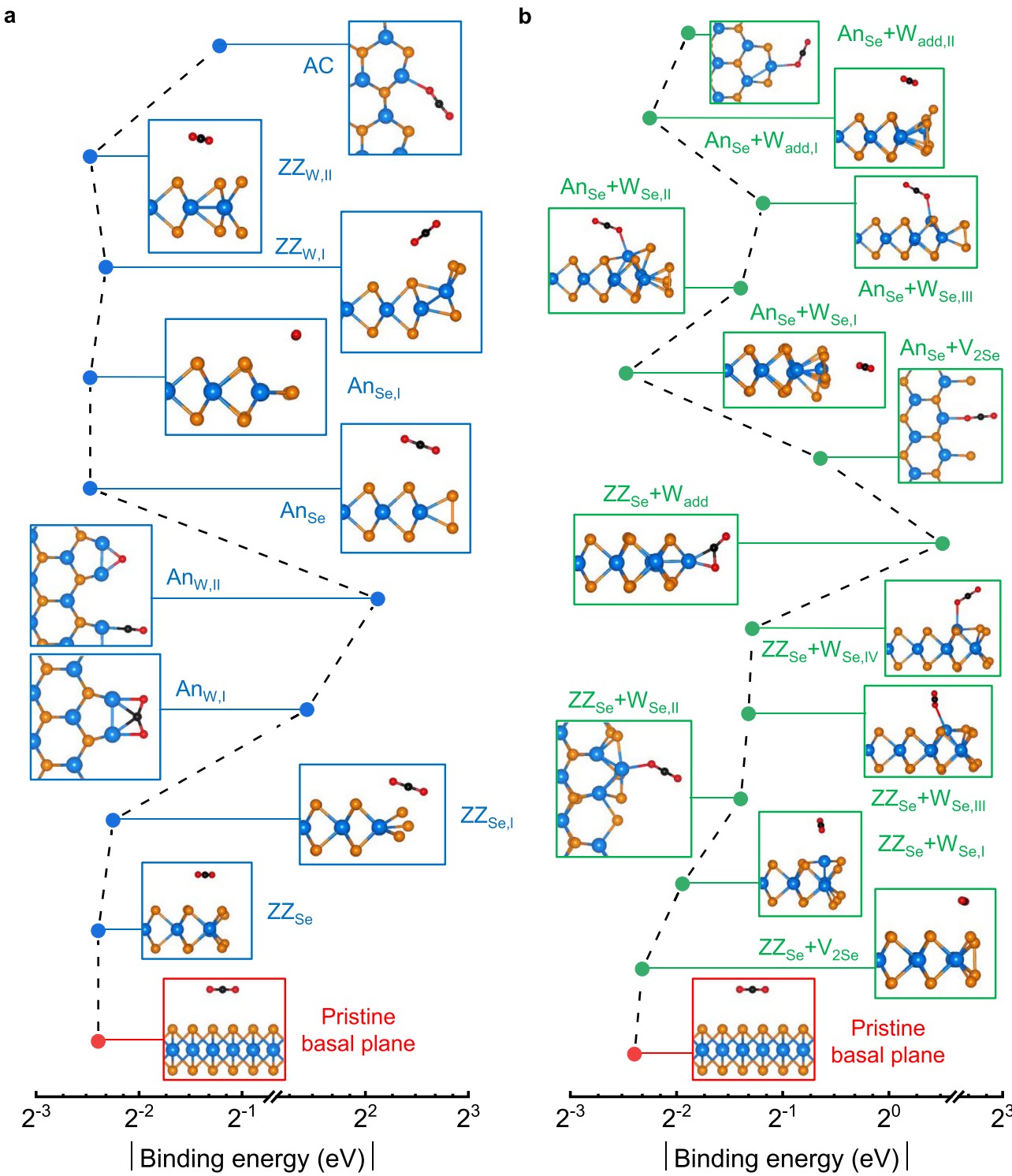

**Fig. 3 CO₂ adsorption. a, b** Absolute value of the binding energy ($\left|E_b^{CO_2}\right|$) and calculation-based relaxed configurations of $CO_2$ at the regular edge (blue boxes) and edge defects (green boxes), respectively. $\left|E_b^{CO_2}\right|$ and relaxed configurations of $CO_2$ on the basal plane (red boxes) are added as the reference. Larger $\left|E_b^{CO_2}\right|$ means stronger $CO_2$ adsorption. Blue- and dark orange- and black- and red-filled circles stand for W, Se, C, and O atoms, respectively. The subscripts Se/W of ZZ/An show the terminated atoms. The dashed line is added as a guide to the eye.

to the contribution of the basal plane and edge, respectively (Supplementary Fig. 28). These fitting factors were obtained by fitting the IQE, which is independent of $S_t$: i.e., $IQE = \gamma(f_b + \frac{f_e}{\bar{P}})$, where $\gamma$ is a constant. Figure 4g shows that the calculated IQE reaches a maximum of 0.23% for the smallest perimeter and the reciprocal fitted curve. This model predicts an IQE ~2.5% for the

ML WSe₂ artificial leaves with a perimeter of 100 nm (albeit by ignoring the effect of the lateral confinement on the electronic structure)[48], which is much larger than that of the highest-productivity plants that have a typical efficiency of ~1% annually[49] or in order of the microalgae grown in bioreactors with ~3% annual yields[50]. Further, the contribution of the edge to the total product varies from 55% to 95% (for $\bar{P}$ from 25 to

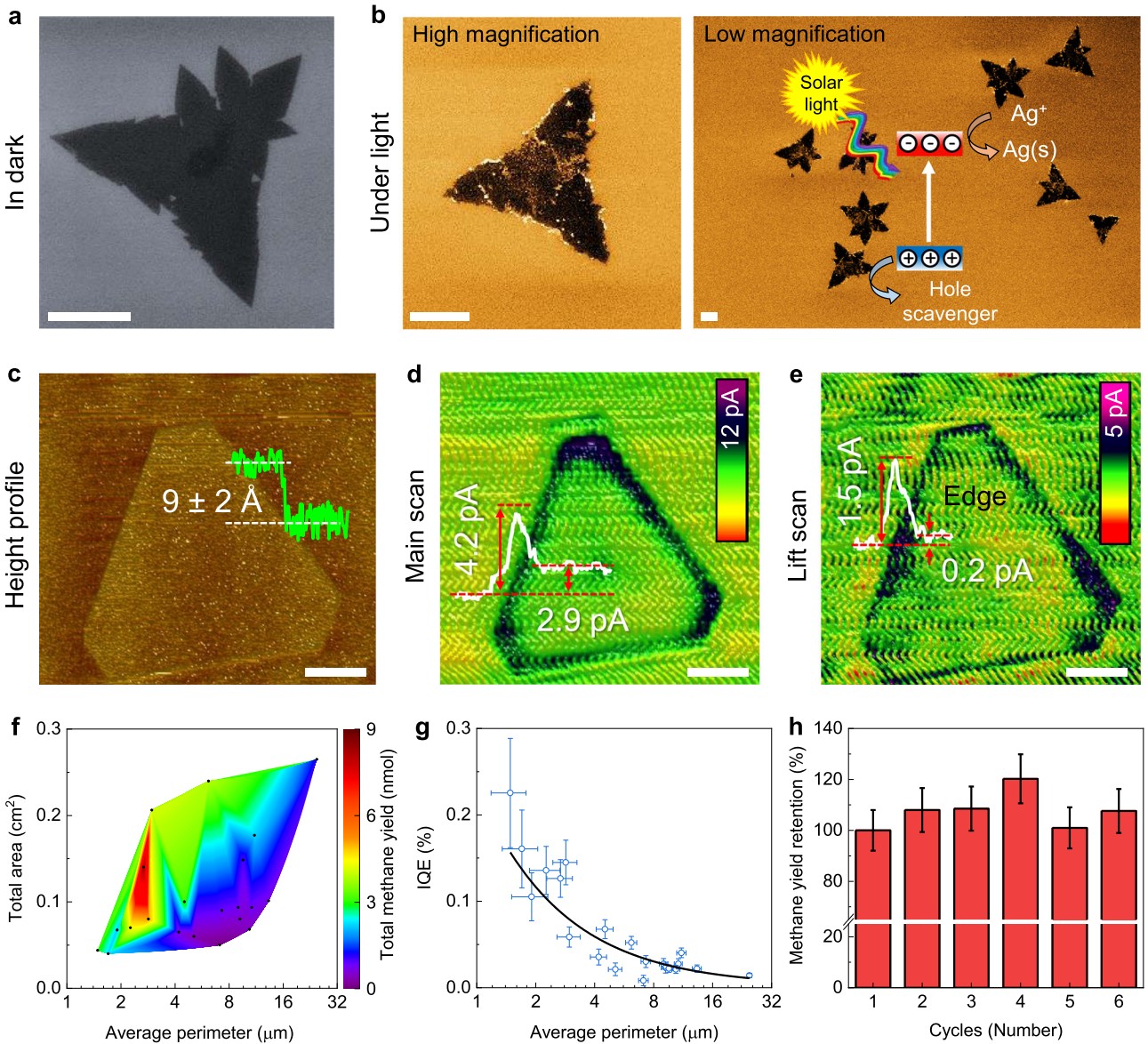

**Fig. 4 Nanoscale redox mapping and PC performance. a** FE-SEM image of the ML WSe$_2$ in dark (control experiment) in the solution containing Ag ions. **b** FE-SEM images of the ML WSe$_2$ under light after Ag photodeposition for 1 h, respectively. Bright regions show the presence of Ag nanoparticles. Inset illustrates the photoreduction mechanism. Scale bar = 2 μm. **c–e** AFM height profile measured in the liquid environment, background normalized SECM feedbacks maps for main, and lift scans, respectively. Scale bar = 1 μm. **f** Color map of the blank-corrected total methane yield as a function of flake sizes (in perimeters) and areas. **g** Blank-corrected IQE as a function of the average flake perimeter. The black line shows the fitted reciprocal curve. **h** Stability test for six cycles. Irradiation time for each cycle is 4 h.

1.5 μm, respectively), confirming that the edge atoms are highly efficient catalytic sites for PC CO$_2$RR. Finally, we calculated the consumed electron rate, $R_e^{edge} = 3.8 \pm 0.7 \, e^- \, s^{-1}$ per edge atom, of the investigated wireless system to compare the intrinsic activities over TMDC materials (Supplementary Table 2 compares our result with other reports). Notably, PC results show $R_e^{basal} = 0.027 \pm 0.004 \, e^- \, s^{-1}$ per in-plane intrinsic defect that is two orders of magnitude smaller than the edge contribution. This illustrates that the reconstructed edges are the dominant active sites for CO$_2$RR. Finally, ML WSe$_2$ flakes displayed a negligible fluctuation after six cycles with the accumulation of 24 h under light irradiation (Fig. 4h), illustrating their long-term PC stability.

In summary, we reported uniformly distributed ML 2H-WSe$_2$ flakes with controlled perimeters to study the contribution of the edge in PC CO$_2$RR. Generally, intrinsic defects on the basal plane and reconstructed edge atoms, that we have theoretically

predicted and experimentally verified, are potential active sites for a PC reaction. The results of the DFT calculations revealed that the basal plane of the ML WSe$_2$ is almost inert to bind CO$_2$ molecules. Nevertheless, AC, An$_W$, and most of the commonly observed defects at the edge can chemisorb linear and bent CO$_2$ molecules. Further, the electronic structure of the edge enabled a better charge transport than the basal plane for redox reactions and prolonged photogenerated carrier lifetimes that are needed for an efficient photocatalyst. Finally, we established that the observed IQE of the wireless PC system is a reciprocal function of the perimeter of the flakes. This size-dependent performance implies that the reconstructed edge of the ML WSe$_2$ plays a critical role in photogenerated charge separation, CO$_2$ adsorption, and charge transfer to the adsorbed reactant. About 95% of the contribution comes from the edges when the characteristic size of the flakes is smaller than 1 μm. Further, the high rate of the

number of electrons consumed per edge atom ($3.8 \pm 0.7\ e^-\ s^{-1}$) is on par with the values for the best-reported wired TMDC electrocatalyst systems, and it can be substantially increased if normalized to the exposed W-terminated sites.

Broadly, this work clarifies that the novel engineered edge of the TMDC materials can be a key factor affecting photocatalysis. The insights attained in this study lead to a deeper understanding of the contribution of the reconstructed edge and suggest that it can act as a high-performance co-catalyst for (photo)electro-chemical applications. Moreover, nanostructures, such as sub-micron scale or quantum dot TMDC flakes, with a preferred edge and number of layers, could be ideal non-precious co-catalysts for wired or wireless HER or $CO_2$RR. Our work thus has wide implications for reactions besides gas-phase PC $CO_2$RR and materials besides $WSe_2$.

## Methods

**Growth of ML $WSe_2$.** To grow $WSe_2$ flakes, a horizontal tube furnace (heating zone length of ~36 cm), equipped with a 1-inch quartz tube, was used. As the precursor, 250 mg of $WSe_2$ powder (99.8%; Alfa Aesar) was added to a semi-cylindrical alumina crucible. A $SiO_2$(300 nm)/Si substrate, with a length and width of 1.7 and 1.5 cm, respectively, was placed facing up at a distance of 15.0 cm downstream from the furnace center. The $WSe_2$ powder was heated up to 950 °C with a subsequent holding time of 5 min before it was cooled to room temperature. See Supplementary Note 1 for details about the growth process and additional information on the ML flakes. Moreover, the poly(methyl methacrylate) (PMMA)-assisted wet-transfer method was applied to transfer the as-grown ML flakes onto arbitrary substrates (Supplementary Note 2).

**Measurements.** Morphologies of ML $WSe_2$ flakes were investigated by using an optical microscope (OM; Olympus BX53M) and also a field-emission scanning electron microscope (FESEM; JEOL-6700F) operated at 6 kV with a working distance of 6 mm. The thickness of the $WSe_2$ flake was measured by using an AFM (Bruker AXS) in the non-contact mode with an arrow-type silicon AFM probe less than 16 nm in diameter (NanoWorld; NCHR-50) and was controlled by a feedback mechanism. The cantilever was driven under a resonant frequency of ~330 kHz and 42 N m$^{-1}$ spring constant. An XPS (VG Scientific ESCALAB 250) system was used to determine the valence states and chemical compositions of the as-grown samples. An HRTEM (JEOL-2100) and Cs-corrected field-emission TEM (JEM–ARM200FTH), operated at 100 kV and 80 kV, respectively, with an acquisition time of 5 s, were used to record the microstructures and SAED patterns. HRTEM images were analyzed by using the Gatan microscopy suite (GMS3) software. PL and Raman spectra were collected on confocal NTEGRA Spectra (NT-MDT) and HORIBA (iHR550) systems using blue (473 nm), red (632 nm), green (532 nm), and violet (405 nm) lasers. The absorption spectrum was measured using a double-beam spectrophotometer (Jasco V-670). See Supplementary Note 3 for details about analyzing OM images, fitting XPS spectra, PL and Raman experiments, and calculation of the overall absorption percentage of ML $WSe_2$.

**Calculations.** The purpose of our DFT calculations is to determine the stable geometries of our ML $WSe_2$ flakes and the DOS at the electronic ground state. Based on those properties, we further determined the active sites for $CO_2$ adsorption without the interaction with light, which is regarded as the first step to understanding $CO_2$ activation and conversion. All the DFT results here were carried out for the electronic ground state. The Vienna ab initio Simulation Package (VASP)[51] was used to perform spin-polarized DFT calculations. Ground state geometries were optimized at the GGA-PBE level[52] with the dispersion-correction method[53]. The cutoff energy of the plane-wave basis was set to 500 eV, and the convergence criterion for residual forces was smaller than 0.01 eV Å$^{-1}$. Basal planes and edges of $WSe_2$ nanoflakes were simulated using a monolayer model with a vacuum space (>20 Å) in the $z$-direction and a nanoribbon model with two vacuum spaces (each >20 Å) in the $y$- and $z$-direction, respectively. $\Gamma$-center of $6 \times 6 \times 1$ mesh was used for sampling the Brillouin zone of the monolayer model and $3 \times 1 \times 1$ mesh was used for the nanoribbon model. For the DOS calculations, since the GGA is well-known for underestimating bandgaps of semiconductors, we used the HSE06 hybrid functional[54] to perform the static electronic calculation based on the optimized structure at the GGA-PBE level. See Supplementary Note 4 for details about the models, $CO_2$ adsorption, and calculation of formation energy.

**Ag photodeposition and nanoscale redox mapping.** The details of the Ag photodeposition are explained in Supplementary Note 5. The SECM experiments were carried out with a Bruker Dimension Icon, in a PeakForce SECM module equipped with a CHI760D electrochemical analyzer and a commercialized nanoelectrode probe (Bruker) operating at −0.4 V versus the Ag pseudo-reference electrodes[55,56]. The solution contained 10 mM hexaammineruthenium (iii) chloride ([Ru(NH$_3$)$_6$]Cl$_3$) and 0.1 M potassium nitrate (KNO$_3$) as a reversible redox mediator and a supporting electrolyte, respectively. See Supplementary Note 5 for details of the SECM setup.

**PC $CO_2$RR.** PC $CO_2$RR experiments were performed using a home-built 7.0 ml stainless steel reactor containing 25 μL DI water at room temperature. The reaction time was 4 h under 1 sun irradiance. The $CH_4$ signal was recorded by injecting 1 ml (Trajan SGE syringe) of the products into a gas chromatography system (GC; Agilent 6890 GC) equipped with a flame ionization detector (FID) and oven operating at 250 °C and 160 °C, respectively. All the PC experiments were performed using ML $WSe_2$ on the $SiO_2$(300 nm)/Si substrate. See Supplementary Note 6 for details about the experimental setup, calculation of the internal quantum efficiency, and the consumed electron rate per active site.

## Data availability

All data are available within the Article and Supplementary Files. Supplementary Dataset 1 presents the optimized structures in VASP and XYZ formats. Supplementary Dataset 2 provides the raw datasets related to the current work. The remaining data that support the findings of this study are available from the corresponding author upon reasonable request.

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

## Acknowledgements

Financial supports provided by the Ministry of Science and Technology (MOST) in Taiwan under the Science Vanguard Project (108-2119-M-002-030, 109-2123-M-002-004, and 110-2123-M-002-006), the i-MATE program of Academia Sinica (AS-iMATE-108-31), as well as the Center of Atomic Initiative for New Materials (AI-Mat), National Taiwan University, and the Featured Areas Research Center Program within the framework of the Higher Education Sprout Project by the Ministry of Education (MOE) in Taiwan (108L9008, 109L9008, and 110L9008) are acknowledged. In addition, we are grateful to the Computer and Information Networking Center, National Taiwan University for the support of high-performance computing facilities. Technical support from Nano-Core, the Core Facilities for Nanoscience and Nanotechnology at Academia Sinica in Taiwan is also acknowledged.

## Author contributions

M.Q. conceived the idea of the study, designed the experiments, prepared the samples, analyzed the results, and wrote the manuscript. A.S. performed the scanning and transmission electron microscopy characterizations. M.Q. and S.Q. carried out the Raman, PL, and TRPL experiments. Y.-R.L. and M.H. performed the DFT simulations. Y.-R.L. wrote the computational method. S.K. and M.Q. carried out the AFM-SECM. C.-Y.H. recorded the XPS spectra. M.Q. performed the GC experiments. I.S. contributed to the design of GC experiments and Ag photodeposition. M.Q. and Y.-F.H. contributed to the design of the CVD system. K.-H.C. and L.-C.C. contributed to directing the research, conceiving the idea of the study, and revising the manuscript. All authors discussed the results and commented on the manuscript.

## Competing interests

The authors declare no competing interests.
