## [Peer review file · Nature Communications]

REVIEWER COMMENTS

Reviewer #1 (Remarks to the Author):

This manuscript reports the synthesis of monolayer (ML) 2H-WSe₂ by a low-pressure vapor deposition approach. The HRTEM and atomic-resolution HAADF-STEM images of the as-prepared ML WSe₂ indicate the presence of zigzag (ZZ), antenna (An) and armchair (AC) configurations together with W adatom at the edge. These images also confirm the existence of comprehensive strain owing to the structure reconstructions vertical to the edge, together with presence of defects, e.g., Se vacancy and W vacancy. Theoretical calculations indicate that the imperfect edge facilitates the adsorption of CO₂ to form linear and bent molecules. And the nanoscale redox mapping corroborates that the edge is the most preferred region for charge transport. This ML WSe₂ shows photocatalytic CO₂ reduction activity to produce CH₄ under one sun irradiation. This work is novel and interesting. I recommend the considering of its publication after addressing the following issues:

1. The authors didn't do any blank experiments, e.g., in the absence of light, photocatalysts or CO₂, in photocatalytic CO₂ reduction.
2. The authors claim that CH₄ is the only product from CO₂ reduction. Why there is no any other products, such as CO? How could WSe₂ achieve such a high selectivity?
3. How is the light absorption ability for ML WSe₂?
4. Only HRTEM and atomic-resolution HAADF-STEM images confirm the presence of vacancies. However, this technique is focused on local area. Can the authors provide the other techniques to confirm the general presence of vacancies in ML WSe₂?
5. It is better to provide in-situ FTIR characterization to study the intermediates of the CO₂-to-CH₄ conversion on ML WSe₂.
6. The edge sites are very important for photocatalytic CO₂ reduction. Why didn't the authors synthesize nano-sized ML WSe₂ instead?
7. How about the activity and selectivity of ML WSe₂ if loaded with various co-catalysts, e.g., Pt, Au and Cu?

Reviewer #2 (Remarks to the Author):

Qorbani et al. present an extremely detailed study on atomically thin flakes of WSe₂ with different sizes. A large array of chemical and physical methods is used to characterize the possible active sites, including

DFT calculations of the structure and the adsorption of CO₂. The materials are tested in photocatalytic CO₂ reduction. High activity for CH₄ production is observed. This can be attributed to reconstructed edge atoms.

Considering the synthesis and characterization of the material, the study of the authors is not only sound and reliable, it is even maximally detailed. I consider the results fully credible. This is not necessarily the case for the photocatalytic results, as will be detailed below. It has not been proven that product formation actually involves CO₂ as reactant. Furthermore, a few other issues need clarification. In summary, a major revision is suggested.

(1) The authors did not prove that CO₂ is the carbon source of CH₄ formation. Seminal works by Guido Mul (~2010) and other authors (Jennifer Strunk, Elena Selli, Adriana Zaleska) have demonstrated that carbon-containing impurities in reactor and/or catalyst are often reacted photocatalytically, whereby products are formed that are identical to CO₂ reduction products. This causes the yield to be dramatically overestimated. Two routes are usually used to prove product formation from CO₂: (i) Use of ¹³CO₂ as isotope label to find ¹³CH₄, or (ii) conducting a blank experiment in absence of CO₂ under otherwise identical conditions. In case of (ii), the amount of CH₄ measured in such a blank experiment must be subtracted from the product yield in the reaction. Either of the two measurements must be performed before the photocatalytic results can be considered reliable.

(2) What happens to the photogenerated holes? Only the reduction product is observed, so there must be an unidentified oxidation product.

(3) Why do the Raman spectra in Figures S4 and S5 look so different? The spectra in Figure S4 should correspond to at least one of lines in each graph of Figure S5. But the noise level and peak height is very different in both Figures. Furthermore, which laser power has been used for the measurements displayed in Figure S4?

(4) Minor issue: In Note 1, Section 1-2, it is not clear which carrier gas (Ar or H₂) was used for which component.

Reviewer #3 (Remarks to the Author):

This paper by Qorbani and co-Authors described a photoactive material based on WSe₂ flakes and a long series of characterizations and calculations aimed at demonstrating the nature of the active sites and the fact that the edges rather than the basal planes.

The paper is original and well carried out, facing the problem from different points of view (physico-chemical characterizations, calculations, fine morphological studies, photochemistry). However, I have a few issues that should be considered:

-Are these really monolayers? Where is the proof for that? Most of text discusses “flakes”, that is a more general term.

-The AFM-SECM explanation makes sense but is based on a qualitative explanation that is proposed by the Authors. I suggest that specific simulations on this system are carried out to reinforce the interpretation of the experimental results.

-Is CH₄ the only CO₂RR product or there are other ones? This is quite surprising and is not discussed in the text

-What is the oxidation reaction that balances the reduction of CO₂?

-The defective edge sites can also act as recombination centers for photogenerated charges. Are there other factors that might influence the edge activity? The longitudinal conductivity vs the “across” one? A higher density of photogenerated charges at the edges rather than at the basal plane (as it is well known for centuries in electrostatics)? I think the Authors should interrogate on these points as well.

Minor points:

Fig1a should be commented in more detail. Moreover, this is an experimental result and it should not be commented in the introduction.

Fig 3 should be vertical: it is very hard to read it in its present form

Fig 4 should be rearranged: both part a) and b) include three images each. Each image should be commented, but this doesn't happen in the manuscript. For example, part b) finds a comment only for the middle and right one, while the left is not discussed.

Reviewer #4 (Remarks to the Author):

In this paper, the authors report on monolayer WSe₂ flakes photocatalyzing the reduction of CO₂ to CH₄. The authors prepared differently sized artificial leaves, performed experimental tests and supplemented them with first principles calculations. The authors observed that intrinsic defects, reconstructed configurations and structural edge defects favourably affect the activity of the leaves.

In this review, I focus on the DFT calculations. The authors claim that regular and imperfect edge configurations adsorb CO₂ more efficiently than in-plane point defects.

DFT calculations were performed at the GGA level with dispersion-correction methods (D3), which is a routinely used well tested approach. From the SI, I see that they used the PAW method and the PBE functional. The cutoff of 500 eV, residual force threshold and other technicalities confirm that these are the garden variety DFT calculations. As such, we can trust the calculations included.

Of course, since the authors did not perform any heavy calculations (such as transition state searching or similar), I see no reason why they would stick to such computationally cheap approaches. PBE has its difficulties, especially for non-metallic systems. While the structures are usually fine, electronic properties – which this paper reports heavily – can be spectacularly wrong. I would recommend that the authors take the optimised structure from the PBE level and perform single-point (no optimization) calculations at a higher level, for instance HSE06 or any other relevant hybrid functional. Oxides and to some extent sulphides have proven too hard a nut for PBE to crack.

1. I miss more information on the structures used in DFT calculations. Preferably, XYZ files should be provided.
2. Even from Figure S13 it is not quite clear what different labels mean with respect to defects. Also, how were these types of defects chosen and why? Are there others as well?
3. Figure S14 nicely shows the phase diagram depicting stable defects at different Se chemical potentials. However, the paper is a bit too light on background. How was the formation energy calculated – provide an equation. Chemical potential is a rather ‘unintuitive’ quantity, which is for gases usually converted to temperature and pressure. How about for Se? What do the investigated values of selenium chemical potentials compare to?
4. Figure S15, S18: As said in the introduction, I really do not trust DOS at the PBE level for semiconductors. Please re-do at a hybrid level.
5. There seems to be no charge transfer from the surface to the CO₂ molecule. Values for E_{ads} around -0.4 eV hint at physisorption. This is what the authors correctly identified in the manuscript. Where is the molecule then activated?
6. In the manuscript, it is not really clear what different terminations, edges etc. represent and which geometries appear in the DFT calculations. Please provide a clearer way of showing that, maybe through a table (can be in the SI).

Important: The authors should make it clear that they made no attempt to capture the excited states and photocatalysis in their DFT calculations. This is acceptable since DFT is a tool for describing ground states (and is occasionally “misused” to calculate higher states by promoting some electrons) but it should be acknowledged. Limitations of such calculations must be discussed.

If the authors could provide a little bit more information on the calculations, did single point calculations at a hybrid level at least for the electronic properties (adsorption energies and geometries are probably

fine at the PBE level) and show the structures explicitly (XYZ files), I would have no problem with the DFT part of the paper. A revision is recommended.

RESPONSE TO THE REVIEWERS' COMMENTS

We thank the reviewers for taking the time to review our manuscript comprehensively. We have addressed their comments, which we find very relevant and important, and we have modified the paper accordingly.

We provide a point-by-point response (in **black** color) to the reviewers' comments and remarks (in **blue** color) below and explain the corresponding changes and additions that we have made to properly address them in the revised version of the manuscript and Supplementary Information.

Fig. R1-1: Photocatalyst yields and blank tests. Total yield for methane (CH_4) and acetaldehyde (CH_3CHO) products after 4 h under one sun irradiation. Our results show that acetaldehyde is a minor product because its yield is in the order of the blank tests.

Changes: The statements about the blank tests have been added to the **Supplementary Notes 6–1** in revised Supplementary Information. **Supplementary Fig. S25** and reference [*Chem. Eur. J.*, **24**, 12739-12746 (2018)] have been added in the revised Supplementary Information. Accordingly, all the reported yields (**Fig. 4f** in the revised manuscript), *IQE*s (**Fig. 4g** in the revised manuscript), the simulated yields of PC CO_2RR (**Supplementary Fig. S28c** and **Supplementary Table S1** in the revised Supplementary Information), and the consumed electron rate per edge atom (**Supplementary Table S2** and **Supplementary Notes 6–3** in the revised Supplementary Information) have been modified all over the revised texts. All the modifications have been highlighted (YELLOW color) in the text.

(2) The authors claim that CH_4 is the only product from CO_2 reduction. Why there is no any other products, such as CO ? How could WSe_2 achieve such a high selectivity?

Response: Thanks for the reviewer’s consideration. As discussed in Comment #1, we have observed the presence of CH_4 and CH_3CHO as the major (selectivity $\geq 90\%$) and minor products, respectively. Regarding the distinguished reviewer’s comment, we have double-checked our products with a helium ionization detector (HID). **Fig. R1-2** shows that CO is not the product of the photocatalytic CO_2 reduction over ML WSe_2 flakes.

Fig. R1-2: Carbon monoxide yield. Signal versus the retention time for the standard gas (black curve), before photocatalytic reaction (blue curve), and after photocatalytic reaction over ML WSe₂ sample (red line). The inset also shows that CO is not the reduction product.

Referring to the recent published papers [*Nat. Energy*, 4, 690-699 (2019) & *Nat. Catal.*, 4, 242-250 (2021) & *Energy Environ. Sci.* 12, 2685-2696 (2019) & *J. Am. Chem. Soc.* 141, 20434-20442 (2019)], the observation of CH₄ implies that CO₂ reduction could be explained by the following reduction reaction pathway: CO₂ + * (forming CO₂* where "*" stands for the active site) following by generation of COOH*, CO* + H₂O, CHO*, CH₂O*, CH₃O*, and CH₄ ↑ + O*, OH*, H₂O + * due to the subsequent proton and electron transfer. Accordingly, O₂ is the oxidation product (**Fig. R1-3**) due to the hole and H₂O reaction. The absence of CO product can be due to the higher binding energy of CO to the edge sites, which can keep it a long time for further reduction reactions. Further, it was proved by our DFT calculation that some of the reconstructed configurations at the edge (not all of them) can adsorb CO₂, implying that the linear density of active site at the edge is not high. Therefore, the absence of C₂ products can be due to the larger distance between the active sites at the edge that can prevent the formation of the C–C bond, which is one of the crucial steps for the formation of C₂ products.

Fig. R1-3: Oxidation product. a, O_2 detection from HID for standard gas, before photocatalytic reaction, and after photocatalytic reaction. b, O_2 detection from gas chromatography–mass spectrometry for the blank and sample after the photocatalytic reaction. Both experiments show that the O_2 -to- N_2 ratio increased after the photocatalytic reaction.

Changes: These statements have been added to the revised manuscript (2nd paragraph of section “Nanoscale redox mapping and photocatalytic performance”). **Supplementary Fig. S26** and **Fig. S27** have been added in the revised Supplementary Information. All the modifications have been highlighted (YELLOW color) in the text.

(3) How is the light absorption ability for ML WSe_2 ?

Response: Thanks for the reviewer’s consideration. The light absorption of the ML WSe_2 and irradiance of the Xe lamp spectrum are shown in **Fig. 1c** (in the revised manuscript) and discussed in detail in **Supplementary Notes 3–4** (in the revised Supplementary Information). Our measurements and

calculations revealed that the average absorption of ML WSe₂ is about 7–8% in the visible region that is in agreement with the theoretical prediction [*Phys Rev B*, 90, 205422 (2014)]. Besides, by considering the spectrum of the utilized Xe lamp (AM 1.5G), the overall absorption percentage is estimated at 4.4 ± 0.3 %.

(4) Only HRTEM and atomic-resolution HAADF-STEM images confirm the presence of vacancies. However, this technique is focused on local area. Can the authors provide the other techniques to confirm the general presence of vacancies in ML WSe₂?

Response: Thanks for the reviewer’s consideration. Besides the HRTEM and atomic-resolution HAADF-STEM images, we have performed several experiments to confirm the presence of defects and vacancies in ML WSe₂: (i) The first analysis is the XPS (**Fig. 1b**, in the revised manuscript) which directly shows the ensemble average of the W-to-Se ratio in the ML flakes. It reveals that the ML flakes are slightly Se-rich, i.e. the presence of V_W. The dependence of δ ($\delta \leq 0$ in W_{1+\delta}Se₂) to \bar{P} was assigned to the intrinsic in-plane defects and Se-terminated edge atoms. Notably, the perimeter-to-area ratio, which is proportional to \bar{P}^{-1} , increases at a faster rate than does δ with decreasing \bar{P} , implying that the edge is not fully terminated with Se atoms. (ii) The second experiment is the low-temperature PL (**Fig. 1d** in the revised manuscript and **Supplementary Fig. S11** in the revised Supplementary Information). These spectra show the presence of an extra broad peak (*D*) at around 1.622 ± 0.002 eV with $\Delta E_{X^0,D} \approx 100$ meV and it can be assigned to the convolution of defect-related [*Nano Lett.*, 20, 2544-2550 (2020)]. The existence of mid-gap states due to defects and vacancies has also been suggested by our DFT calculations (**Supplementary Fig. S16**, in the revised Supplementary Information). The *D*-to-*X*⁰ peak ratio has the same trend as we observed in XPS analyses. Regarding the reviewer’s consideration, we have performed the room-temperature PL and Raman scattering of the ML flakes grown by low-pressure vapor deposition (VT-WSe₂) and prepared by a micromechanical exfoliation (ME-WSe₂) (**Fig. R1-4a** and **b**). As shown in **Fig. R1-4c**, the PL intensity of the VT-WSe₂ is much lower than the ME-WSe₂ due to the presence of mid-gap states and a high non-radiative recombination rate. In addition, the high *X*⁻-to-*X*⁰ ratios are 0.46 and 1.25 for the ME-WSe₂ and VT-WSe₂, respectively. This implies that the presence of localized charge carriers is due to the higher defect density in the VT-WSe₂ sample. (iii) The third experiment is the Raman scattering. In contrast to VT-WSe₂, the Raman spectrum of the ME-WSe₂ shows an intense *A*(*M*) peak with a small shoulder at ~250 cm⁻¹ (*E*_{2g}¹ + *A*_{1g}) (**Fig. R1-4d**). We believe the above-mentioned three experiments, beyond the HRTEM and HAADF-STEM, could be enough to quickly confirm the presence of vacancies and defects in the ML WSe₂. However, quantifying the defect density by using the above-mentioned techniques requires a separate set of detailed investigations.

Fig. R1-4: Comparison between ML WSe₂ grown by vapor deposition and prepared by micromechanical exfoliation. **a** and **b**, Optical microscope and AFM height profile images, respectively. **c** and **d**, room-temperature PL and Raman spectra recorded by the red laser (632 nm), respectively.

Changes: Supplementary Fig. S10 has been added in the revised Supplementary Information and commented in the revised manuscript (1st paragraph of section “**Characterizations of monolayer WSe₂ artificial leaf**”). All the modifications have been highlighted (YELLOW color) in the text.

(5) It is better to provide in-situ FTIR characterization to study the intermediates of the CO₂-to-CH₄ conversion on ML WSe₂.

Response: Thanks for the reviewer’s suggestion. For in-situ FTIR, we used to mix a few milligrams of the catalyst with KBr using our homemade system (Bruker Tensor 27 IR spectrometer with MCT detector

[*Chem. Eng. J.*, 430, 132853 (2022)]). Considering the density of the bulk WSe₂ of 9.3 g cm⁻³, the thickness of the ML flakes of ~0.8 nm, and the area of the grown film on the SiO₂ substrate of ~0.2 cm², the mass of the ML catalyst is estimated at ~0.15 μg. The estimated mass is about five orders of magnitude lower than the normal powder systems, resulting in a much lower FTIR signal. It should be noted that, solar fuel formation rate (i.e., non-gravimetric scale) for the hydrothermally derived powder systems, such as TiO₂ [*Small* 14, 1702928 (2018)], graphene oxide [*Nanoscale* 5, 262 (2013) & *Nano Lett.* 14, 6097-6103 (2014)], SnS₂ [*Nat. Commun.* 9, 169 (2018)], and graphitic-C₃N₄ [*Chem. Engineering J.* 430, 132853 (2022)], is in the range of hundred nmol h⁻¹ to a few μmol h⁻¹, which is about two to three orders of magnitude larger than that of thin films (typically, < 10 nmol h⁻¹), e.g., SnS₂ grown by CVD [*Nano Energy*, 72, 104717 (2020)]. In the powder-based systems, such as TiO₂, ZnS/ZnIn₂S₄ and graphitic-C₃N₄, we were able to conduct mechanistic studies using *in situ* FTIR [*ACS Appl. Mater. & Inter.* 11, 25186 (2019); *Nano Energy* <https://doi.org/10.1016/j.nanoen.2021.106809> (2021); *Chem. Eng. J.*, 430, 132853 (2022)]. However, in the present study, the solar fuel formation rate is about 2-3 nmol h⁻¹ (see **Fig. 4f**, in the revised manuscript), which is typical also for other CVD-grown thin films studied in our group. Under such circumstances, our current FTIR system cannot provide a meaningful signal-to-noise ratio to study the intermediates. Our efforts on various TMDC thin films, with even larger thicknesses of a few tens of nanometers, have not resulted in a reportable signal. In the future, by re-designing our FTIR system, we do hope to boost the signal-to-noise ratio, so as to investigate ultrathin films. We would appreciate the distinguished reviewer's understanding of not providing FTIR data at present time.

(6) The edge sites are very important for photocatalytic CO₂ reduction. Why didn't the authors synthesize nano-sized ML WSe₂ instead?

Response: This is a very important point. We have already mentioned this point in section “**Conclusions**” in the first version of the manuscript that nanostructures, such as sub-micron scale or quantum dot TMDC flakes, with a preferred edge and number of layers, could be ideal non-precious co-catalysts for wired or wireless HER or CO₂RR. However, our target was to controllably grow clean ML WSe₂ flakes with different lateral sizes, which can be measured by optical microscope as the large-scale, fast, and non-destructive measurement method (notably, for each sample we have measured the lateral size of ~10³ flakes using the optical microscope images). Regarding the resolution of the optical microscope of ~0.3 μm, the smallest average lateral size of $\bar{L} \approx 0.5 \mu\text{m}$ (or $\bar{P} = 1.5 \mu\text{m}$) was selected as our lower limit. Otherwise, the error of measuring the exact lateral sizes and the effective area will be too high. On the other hand, we have tried to prevent any additional complexity in this study, i.e., the lateral confinement. Lateral confinement can change the electronic structure of the ML TMDC [*Appl. Phys. Lett.*, 106, 233113 (2015)], which can have a large impact on photocatalytic performance. As discussed in

the last paragraph of the revised manuscript, our model predicts that nano-sized ML WSe₂ could reach an IQE over 2.5% for the average lateral size of $\bar{L} \approx 30$ nm (or $\bar{P} \approx 100$ nm) by ignoring the effect of the lateral confinement. However, we have recently started to use a top-down method to synthesize ML TMDC quantum dots with different lateral sizes following reference [*Angew. Chem. Int. Ed. Engl.*, 54, 5425-5428 (2015)] to study the lateral confinement effect on the electronic structure, photocatalytic performance, and selectivity of the final products. Here, our bottom-up growth method, i.e., low-pressure vapor deposition, is an appropriate way to produce a uniform film in terms of the number of layers, lateral sizes and distribution, allowing us to perform comparative study on the contribution from edge and basal plane. The reviewer's excellent suggestion merits a separate study, but is beyond the scope of the present study. Actually, instead of a single-phase nano-sized catalyst, we are going to investigate the role of nano-sized few-layer WSe₂ as a non-precious co-catalyst decorating on the well-known absorbers such as C-SnS₂ [*Nat. Commun.*, 9, 169 (2018)], Cu₂O [*Nat. Energy*, 4, 957-968 (2019)], and blue TiO₂ [*Energ. Environ. Sci.*, 11, 3183-3193 (2018)] in the future.

Changes: Reference [*Appl. Phys. Lett.*, 106, 233113 (2015)] and a statement about the effect of lateral confinement have been added in in the revised manuscript.

(7) How about the activity and selectivity of ML WSe₂ if loaded with various co-catalysts, e.g., Pt, Au and Cu?

Response: Thanks for the reviewer's suggestion. As discussed in the introduction section of the manuscript, using co-catalysts (like precious Pt) results in a conversion rate that is higher than that of other strategies by one to two orders of magnitude (see **Supplementary Table S1** in the revised Supplementary Information). We agree that different co-catalysts can also change the selectivity of the final products. In general, co-catalysts play four important roles: (i) boosting charge separation/transfer, (ii) improving the activity and selectivity of CO₂ reduction, (iii) enhancing the stability of photocatalysts, and (iv) suppressing side or back reactions as discussed in reference [*Adv. Mater.*, 30, 1704649 (2018)]. Nevertheless, this study mainly aimed for shedding light on the highly active reconstructed edges and studying the size-dependent photocatalytic activity of ML WSe₂ photocatalysts.

Generally, co-catalysts could be decorated on the basal plane or at the edge sites of the ML TMDC. They can easily be dominant sites for CO₂ reduction and suppress the role of the edge. Based on the distinguished reviewer's comment, we have performed a sputtering process to load Pt co-catalyst on ML WSe₂ followed by performing a photocatalytic CO₂ reduction test. As shown in **Fig. R1-5**, not only the photocatalytic performance but also the selectivity of the products changed after the introduction of Pt co-catalyst. The reduction of the total yield of CH₄ can be due to the coverage of the edge by Pt co-catalyst, as the main active site for photoreduction of CO₂ to CH₄. Moreover, the emerging of a C₂ product can be

assigned to the presence of Pt co-catalyst as reported by others [*Energy Environ. Sci.* 12, 2685-2696 (2019)]. However, it still needs several blank tests to prove the presence of other products, which is not in the scope of our current research.

Fig. R1-5: The role of Pt co-catalyst on the photocatalytic CO₂ reduction. Total yield for methane (CH₄) and acetaldehyde (CH₃CHO) products after 4 h under one sun irradiation. The average perimeter of the ML WSe₂ is ~10 μm.

Further, controlling the size, loading amount, uniformity, and decorated place (basal plane or edge) of the co-catalyst requires a separate set of detailed investigations. Another challenge is the synthesis of co-catalyst on the ML TMDC flakes prepared by a bottom-up vapor deposition method without immersing the ML flakes inside an aqueous liquid, which can quickly damage them, forming a crumbled flake after drying (**Fig. R1-6**), or detach most of the flakes. As discussed in Comment #6, we are investigating the role of nano-sized TMDC materials as non-precious co-catalyst or decorated with noble metal co-catalysts. We hope that the distinguished reviewer will find our response to his/her comment satisfactory.

Fig. R1-6: Crumbled flake. AFM height profile of a dried flake after immersing in deionized water.

Therefore, regarding the reviewer's suggestion, we have performed the blank tests (**Fig. R2-2**) [*Chem. Eur. J.*, 24, 12739-12746 (2018)]: (i) With WSe₂ photocatalyst/with CO₂/without light (with the production of Y_{b1}), (ii) With WSe₂ photocatalyst/without CO₂/with light (with the production of Y_{b2}), and (iii) Without WSe₂ photocatalyst/with CO₂/with light (with the production of Y_{b3}). Finally, the values of products measured in the blank experiments are subtracted from the product yield in the photocatalytic reaction as,

$$Y = Y_{exp} - Y_{b1} - Y_{b2} - Y_{b3}. \quad (1-1)$$

where Y is the total blank-corrected yield. Regarding these corrections, the total methane yields (**Fig. 4f** in the revised manuscript) and *IQEs* (**Fig. 4g** in the revised manuscript) have been modified (Notably the highest *IQE* value decreased from 0.29% to 0.23% after the subtraction of the blank test). Then, we have re-fitted the *IQE* plot with $f_e = 0.23$ and $f_b = 0.007 f_e$, showing that the activity of the basal plane is in the order of the blank experiments. It is in well-agreement with our DFT calculations and SECM results that reveal the basal plane of the ML TMDCs is inert, which is consistent with the other reports [*Science*, 317, 100-102 (2007) & *Nat. Mater.*, 15, 48-53 (2016) & *Nat. Commun.*, 8, 15113 (2017)]. Therefore, the new fitting of the *IQE* has resulted in a higher contribution for the edge sites (more than two orders of magnitude compared with the basal plane) in the photocatalytic activity of the ML WSe₂. Moreover, we have re-calculated the consumed electron rate of $3.8 \pm 0.7 e^- s^{-1}$ per edge atom.

Fig. R2-2: Photocatalyst yields and blank tests. Total yield for methane (CH₄) and acetaldehyde (CH₃CHO) products after 4 h under one sun irradiation. Our results show that acetaldehyde is a minor product because its yield is in the order of the blank tests.

Changes: The statements about the blank tests have been added to the **Supplementary Notes 6–1** in revised Supplementary Information. **Supplementary Fig. S25, Supplementary Fig. N8**, and references [*J. Phys. Chem. A* 119, 4658-4666 (2015) & *J. Phys. Chem. C* 124, 10981-10992 (2020) & *Chem. Eur. J.*, 24, 12739-12746 (2018)] have been added in the revised Supplementary Information. Accordingly, all the reported yields (**Fig. 4f** in the revised manuscript), *IQEs* (**Fig. 4g** in the revised manuscript), the simulated yields of PC CO₂RR (**Supplementary Fig. S28c** and **Supplementary Table S1** in the revised Supplementary Information), and the consumed electron rate per edge atom (**Supplementary Table S2** and **Supplementary Notes 6–3** in the revised Supplementary Information) have been modified all over the revised texts. All the modifications have been highlighted (YELLOW color) in the text.

(2) What happens to the photogenerated holes? Only the reduction product is observed, so there must be an unidentified oxidation product.

Response: We thank the reviewer very much for pointing this out. Regarding the distinguished reviewer's comment, we have qualitatively measured the photocatalytic products by using a helium ionization detector (HDI; **Fig. R2-3a**) and gas chromatography–mass spectrometry (GC-MS; **Fig. R2-3b**). Both of these experiments show that the O₂-to-N₂ ratio has increased after the photocatalytic CO₂ reduction. Therefore, O₂ is the oxidation product due to the reaction of photogenerated holes and H₂O through the following equation [*Chem. Eng. J.*, 430, 132853 (2022) & *Nat. Energy*, 4, 690-699 (2019)]:

Fig. R2-3: Oxidation product. a, O_2 detection from HID for standard gas, before photocatalytic reaction, and after photocatalytic reaction. b, O_2 detection from gas chromatography–mass spectrometry for the blank and sample after the photocatalytic reaction. Both experiments show that the O_2 -to- N_2 ratio increased after the photocatalytic reaction.

Changes: The statement, about the oxidation reaction (with reference [*Chem. Eng. J.*, 430, 132853 (2022)]), and **Supplementary Fig. S27** have been added and highlighted in the revised manuscript (2nd paragraph of section “Nanoscale redox mapping and photocatalytic performance”) and Supplementary Information, respectively.

(3) Why do the Raman spectra in Figures S4 and S5 look so different? The spectra in Figure S4 should correspond to at least one of lines in each graph of Figure S5. But the noise level and peak height is very different in both Figures. Furthermore, which laser power has been used for the measurements displayed in Figure S4?

Response: Thanks for the reviewer's consideration and point. We are sorry for the confusion. The intensity of the vibrational mode depends on the excitation wavelength [*ACS Nano* 8, 9629-9635 (2014)]. Therefore, we have used different excitation wavelengths to characterize the grown flakes with more details. **Supplementary Fig. S4** and **Fig. S5** (in the revised Supplementary Information) present the Raman spectra of the samples using laser excitations of 632 nm (red; laser power ~15 mW) and 532 nm (green; different laser power from 0.5 to 15 mW), respectively. Further, **Supplementary Fig. S3** compares the Raman spectra of the ML WSe₂ using different laser excitation wavelengths of 473, 532, and 632 nm.

Changes: Laser powers have been added and highlighted in **Supplementary Notes 3–3**, and the caption of **Supplementary Fig. S4** and **Fig. S5** in the revised Supplementary Information.

(4) Minor issue: In Note 1, Section 1-2, it is not clear which carrier gas (Ar or H₂) was used for which component.

Response: Thanks for the reviewer's point. Both Ar and H₂ were introduced as the carrier gases during the chemical vapor deposition process, simultaneously.

Changes: This statement has been clarified and highlighted in **Supplementary Notes 1–2** (in the revised Supplementary Information).

Then, we have used PL measurement to check the uniformity of the number of layers after the growth process. **Fig. R3-2** presents the PL spectra of the various flakes on different samples. All these spectra show that the optical band gap of ~ 1.68 eV, with an error of ≤ 0.01 eV. Regarding their intense intensity and high band gap as compared with BL flakes, one can conclude that the number of layers is uniform (i.e., ML) in each sample.

Fig. R3-2: Uniformity of the number of layers by photoluminescence. **a** to **c**, the PL spectra of the various flakes at different random places in each sample with average flake sizes of $L \approx 0.7 \mu\text{m}$, $L \approx 1.8 \mu\text{m}$, and $L \approx 4.2 \mu\text{m}$, respectively. Notably, the variation of the PL intensities is less than 20% for each panel.

Occasionally, the presence of BL regions on some of the large flakes was observed, mostly close to the nucleation sites (**Fig. R3-3a**). Our investigations show that rather than initial nucleation sites, secondary nucleation sites can also be the origin of the growth of the second layers. AFM and HRTEM images show that the apexes of some flakes are the secondary nucleation sites (**Fig. R3-3b** and **c**). Notably, the density of the BL region is much lower in the smaller flakes due to the less feeding rate during the growth process. **Fig. R3-3d** and **e** reveal that the second layer is already twisted to a small angle due to an energetically favorable stacking structure, displaying moiré patterns. So, the ensemble averaging estimates a few percent of BL flakes in each sample.

Fig. R3-3: Growth of the partial second layer. **a**, Optical microscope images of flakes. It should be noted that the ML and BL regions can be distinguished quickly by their different contrasts **b**, AFM height profiles. **c** and **d**, TEM and HRTEM images. **e**, SAED pattern of the BL region.

Changes: These statements and **Supplementary Note Fig. N2, Fig. N3, Fig. N4** have been added and highlighted in **Supplementary Notes 1–1** (in the revised Supplementary Information). All the corrections have been highlighted (YELLOW color) in the text.

(2) The AFM-SECM explanation makes sense but is based on a qualitative explanation that is proposed by the Authors. I suggest that specific simulations on this system are carried out to reinforce the interpretation of the experimental results.

Response: Thanks for the reviewer's consideration. The COMSOL Multiphysics software (Chemical Reaction Engineering Module; COMSOL, Inc., Burlington, MA) is the best one for solving the diffusion equation (Laplace's equation) and simulating the SECM data by considering required boundary conditions [J. Am. Chem. Soc. 138, 5123-5129 (2016) & Nanotechnology 28, 095711 (2017) & Proc. Natl. Acad. Sci. USA 116, 11618-11623 (2019)]. Although, we do not have access to this software to perform the simulation which requires a separate set of detailed investigations. Therefore, we have utilized the current models and equations to reinforce the interpretation of the AFM-SECM data more in-depth. As shown in **Fig. R3-4**, the SECM normalized responses ($\frac{I}{I_0}$, where I_0 is the steady-state current when the tip is far from the surface) as a function of the normalized tip-surface distance ($\frac{z}{a}$ where a is the radius of the Pt tip) over the WSe₂ and SiO₂/Si substrate have been recorded. It shows that a negative and slightly positive feedback current over SiO₂/Si (~0.75 at $\frac{z}{a} = 1$) substrate and monolayer WSe₂ (~1.05 at $\frac{z}{a} = 1$), respectively. It reveals that there is still slight charge transfer (of course, it is not comparable to the Pt film as shown in the inset of **Fig. R3-4**) over the semiconducting film. An analytical approximation has been proposed for fitting the positive and negative feedback currents on the basal plane of WSe₂ and SiO₂/Si substrate, respectively, as references [Anal. Chem. 61, 1221-1227 (1989) & J. Phys. Chem. B 102, 9946-9951 (1998)]:

$$\frac{I_{basal}^+}{I_0} = K_{1,basal}^+ + \frac{K_{2,basal}^+}{\left(\frac{z}{a}\right)} + K_{3,basal}^+ e^{\frac{K_{4,basal}^+}{\left(\frac{z}{a}\right)}}, \quad (3-1)$$

$$\frac{I^-}{I_0} = \left(K_1^- + \frac{K_2^-}{\left(\frac{z}{a}\right)} + K_3^- e^{\frac{K_4^-}{\left(\frac{z}{a}\right)}} \right)^{-1}, \quad (3-2)$$

where $K_{1,basal}^+$, $K_{2,basal}^+$, $K_{3,basal}^+$, $K_{4,basal}^+$, K_1^- , K_2^- , K_3^- , and K_4^- are dimensionless constants. Our SECM feedback results show that the positive and negative feedback currents are independent of $K_{2,basal}^+$ and K_2^- . So, we have used the following equations to fit the SECM feedback currents (see red and blue curves in **Fig. 4**):

$$\frac{I_{basal}^+}{I_0} = K_1^+ + K_3^+ e^{\frac{K_4^+}{\left(\frac{z}{a}\right)}}, \quad (3-3)$$

$$\frac{I^-}{I_0} = \left(K_1^- + K_3^- e^{\frac{K_4^-}{\left(\frac{z}{a}\right)}} \right)^{-1}, \quad (3-4)$$

with $K_{1,basal}^+$, K_1^- , $K_{3,basal}^+$, K_3^- , $K_{4,basal}^+$, and K_4^- of 1.05, 1.35, -0.063, 0.36, -97.92, -11.48, respectively. Additionally, the equation (3-1) was used to fit the $\frac{I_{edge}^+}{I_0}$ (positive feedback) where $K_{1,edge}^+$, $K_{2,edge}^+$, $K_{3,edge}^+$, and $K_{4,edge}^+$ are about 0.3, 3.2, 0.69, and -0.06, respectively. Such behavior shows that the edge region behaves like a metal albeit with a less charge transfer rate, assigning to the spatial confinement in the across direction (it should be noted that DFT calculations show that the ML WSe₂ displays a semiconducting nature atomically far from the edge region). Given that, the feedback normalized responses reveal the excellent charge transfer properties of the edge compared with the basal plane. We hope that the distinguished reviewer will find our response to his/her comment satisfactory.

Fig. R3-4: SECM feedback mode. SECM normalized response as a function of the tip-surface distance. Red, green, and blue curves represent the fitted curves for positive ($\frac{I_{basal}^+}{I_0}$), positive $\frac{I_{edge}^+}{I_0}$ and negative $\frac{I^-}{I_0}$ feedback currents, respectively. The inset shows the SECM normalized response of the tip-Pt distance.

Changes: These statements have been added to the revised manuscript (1st paragraph of section “Nanoscale redox mapping and photocatalytic performance”). **Supplementary Fig. S24** has been added in the revised Supplementary Information. The above-mentioned equations have been added in **Supplementary Notes 5–2** in the revised Supplementary Information. All the corrections have been highlighted (YELLOW color) in the text.

(3) Is CH₄ the only CO₂RR product or there are other ones? This is quite surprising and is not discussed in the text.

Response: Thanks for the reviewer's consideration. We have observed the presence of CH₄ and CH₃CHO as the major (selectivity $\geq 90\%$) and minor products, respectively. This selectivity have been calculated after performing the following blank experiments (**Fig. R3-5**) [*Chem. Eur. J.*, 24, 12739-12746 (2018)]: (i) With WSe₂ photocatalyst/with CO₂/without light (with the production of Y_{b1}), (ii) With WSe₂ photocatalyst/without CO₂/with light (with the production of Y_{b2}), and (iii) Without WSe₂ photocatalyst/with CO₂/with light (with the production of Y_{b3}). Finally, the values of products measured in the blank experiments have subtracted from the product yield in the photocatalytic reaction as,

$$Y = Y_{exp} - Y_{b1} - Y_{b2} - Y_{b3}. \quad (1-1)$$

where Y is the total blank-corrected yield. Regarding these corrections, the total methane yields (**Fig. 4f** in the revised manuscript) and IQE s (**Fig. 4g** in the revised manuscript) have been modified (Notably the highest IQE value decreased from 0.29% to 0.23% after the subtraction of the blank test). Then, we have re-fitted the IQE plot with $f_e = 0.23$ and $f_b = 0.007 f_e$, showing that the activity of the basal plane is in the order of the blank experiments. It is in well-agreement with our DFT calculations and SECM results that reveal the basal plane of the ML TMDCs is inert, which is consistent with the other reports [*Science*, 317, 100-102 (2007) & *Nat. Mater.*, 15, 48-53 (2016) & *Nat. Commun.*, 8, 15113 (2017)]. Therefore, the new fitting of the IQE has resulted in a higher contribution for the edge sites (more than two orders of magnitude compared with the basal plane) in the photocatalytic activity of the ML WSe₂. Moreover, we have re-calculated the consumed electron rate of $3.8 \pm 0.7 e^- s^{-1}$ per edge atom.

Fig. R3-5: Photocatalyst yields and blank tests. Total yield for methane (CH_4) and acetaldehyde (CH_3CHO) products after 4 h under one sun irradiation. Our results show that acetaldehyde is a minor product because its yield is in the order of the blank tests.

Additionally, regarding the distinguished reviewer's comment, we have double-checked our products with a helium ionization detector (HID). **Fig. R3-6** shows that CO is not the product of photocatalytic CO_2 reduction over ML WSe_2 .

Fig. R3-6: Carbon monoxide yield. Signal versus the retention time for the standard gas (black curve), before photocatalytic reaction (blue curve), and after photocatalytic reaction over ML WSe₂ sample (red line). The inset also shows that CO is not the reduction product.

Referring to the recent published papers [*Nat. Energy*, 4, 690-699 (2019) & *Nat. Catal.*, 4, 242-250 (2021) & *Energy Environ. Sci.* 12, 2685-2696 (2019) & *J. Am. Chem. Soc.* 141, 20434-20442 (2019)], the observation of CH₄ implies that CO₂ reduction could be explained by the following reduction reaction pathway: CO₂ + * (forming CO₂* where "*" stands for the active site) following by generation of COOH*, CO* + H₂O, CHO*, CH₂O*, CH₃O*, and CH₄ ↑ + O*, OH*, H₂O + * due to the subsequent proton and electron transfer. Accordingly, O₂ is the oxidation product (see Comment #4) due to the hole and H₂O reaction. The absence of CO product can be due to the higher binding energy of CO to the edge sites which can keep it a long time for further reduction reaction. It was proved by our DFT calculation that some of the reconstructed configurations at the edge (not all of them) can adsorb CO₂, implying that the linear density of active site at the edge is not high. Therefore, the absence of C₂ products can be due to the larger distance between the active sites at the edge that can prevent the formation of C–C bond, which is one of the crucial steps for the formation of C₂ products.

Changes: The statements about the blank tests have been added to the **Supplementary Notes 6–1** in revised Supplementary Information and revised manuscript (2nd paragraph of section “**Nanoscale redox mapping and photocatalytic performance**”). **Supplementary Fig. S25, Fig. S26**, reference [*Chem. Eur. J.*, 24, 12739-12746 (2018)] have been added in the revised Supplementary Information. Accordingly, all the reported yields (**Fig. 4f** in the revised manuscript), *IQE*s (**Fig. 4g** in the revised

manuscript), the simulated yields of PC CO₂RR (**Supplementary Fig. S28c** and **Supplementary Table S1** in the revised Supplementary Information), and the consumed electron rate per edge atom (**Supplementary Table S2** and **Supplementary Notes 6–3** in the revised Supplementary Information) have been modified all over the revised texts. All the modifications have been highlighted (YELLOW color) in the text.

(4) What is the oxidation reaction that balances the reduction of CO₂?

Response: We thank the reviewer very much for pointing this out. Regarding the distinguished reviewer’s comment, we have qualitatively measured the photocatalytic products by using a helium ionization detector (HDI; **Fig. R3-7a**) and gas chromatography–mass spectrometry (GC-MS; **Fig. R3-7b**). Both of these experiments show that the O₂-to-N₂ ratio has increased after the photocatalytic CO₂ reduction. Therefore, O₂ is the oxidation product due to the reaction of photogenerated holes and H₂O through the following equation [*Chem. Eng. J.*, 430, 132853 (2022) & *Nat Energy*, 4, 690-699 (2019)]:

Fig. R3-7: Oxidation product. **a**, O₂ detection from HID for standard gas, before photocatalytic reaction, and after photocatalytic reaction. **b**, O₂ detection from gas chromatography–mass spectrometry for the

blank and sample after the photocatalytic reaction. Both experiments show that the O₂-to-N₂ ratio increased after the photocatalytic reaction.

Changes: These statements and **Supplementary Fig. S27** have been added and highlighted in the revised manuscript (2nd paragraph of section “**Nanoscale redox mapping and photocatalytic performance**”) and Supplementary Information, respectively. All the modifications have been highlighted (YELLOW color) in the text.

(5) The defective edge sites can also act as recombination centers for photogenerated charges. Are there other factors that might influence the edge activity? The longitudinal conductivity vs the “across” one? A higher density of photogenerated charges at the edges rather than at the basal plane (as it is well known for centuries in electrostatics)? I think the Authors should interrogate on these points as well.

Response: We thank the reviewer very much for pointing this out. Yes, several factors can further influence the edge activity listed below:

(i) DFT calculations have shown the metallic behavior of the edge region (AC and An edges). It means the charge transport across the edge is favorable than the basal plane. The higher current at the edge observed by SECM reflects this point.

(ii) DFT calculations have also suggested that the edge and basal plane can form a local homostructure, which improves charge separation.

(iii) Regarding the electronic density of states and the recombination rate at the edge region: We agree with the reviewer’s point on the possibility of the defective edge sites to act as recombination centers, though we are unable to provide direct evidence for the recombination. Meanwhile, the density of the photogenerated charge carriers and/or subsequent carrier separation could be higher at the reconstructed edge regions, which can enhance the probability of charge transfer to the adsorbed CO₂. All these factors be it positive or negative contribution, are convoluted with each other to influence the overall activity.

(iv) The capability of the edge sites to adsorb and activate CO₂ molecules due to the presence of several structural defects, reconstruction, and imperfect configurations. We have covered this point substantially, as summarized in **Fig. 3** (in the revised manuscript).

Changes: These statements have been discussed, emphasized, and highlighted in the revised manuscript.

Minor points:

Fig 1a should be commented in more detail. Moreover, this is an experimental result and it should not be commented in the introduction.

Fig 3 should be vertical: it is very hard to read it in its present form.

Fig 4 should be rearranged: both part a) and b) include three images each. Each image should be commented, but this doesn’t happen in the manuscript. For example, part b) finds a comment only for the middle and right one, while the left in not discussed.

Response: Thanks for the reviewer's points. We have modified the manuscript as suggested by the distinguished reviewer. All the modifications have been highlighted (YELLOW color) in the text.

Changes: **Fig. 1a** (in the revised manuscript) has been commented in the experimental result. **Fig. 3** (in the revised manuscript) has been replaced with two vertical panels, accordingly. **Fig. 4** (in the revised manuscript) has also been modified and commented in the main text.

Fig. R4-1: Intrinsic point defects by density functional theory calculations. a, Top-view images of the relaxed structure calculated by density functional theory (DFT) of the pristine basal plane and, b to d, Se vacancy, WSe_x vacancy, and antisite families of intrinsic defects, respectively. Blue- and dark orange-filled circles stand for W and Se atoms. Light blue and dashed red circles stand for W and Se deficiencies. White arrows show the antisite defects. The lattice constant was calculated at 3.28 Å for the relaxed pristine ML WSe₂.

As shown in **Supplementary Fig. S13a**, we have used HAADF-STEM images to experimentally characterize the defects on the basal plane of the ML WSe₂. It shows several intrinsic point defects consistent with other reports [*Nano Lett.* 13, 2615-2622 (2013) & *Nat. Commun.* 6, 6293 (2015) & *Phys. Rev. Lett.* 119, 046101 (2017)], including V_{Se} (commonly observed), V_{2Se}, V_{WSe_x}, W_{Se}, and Se_W, that were non-uniformly distributed in the basal plane of ML flakes with a density of $\sim 5 \times 10^{13} \text{ cm}^{-2}$ (i.e., the defect concentration is $\sim 1.5 \text{ at\%}$). For the V_{2Se} defects, we observed several configurations such as V_{Se+Se} I, II, and III as well as V_{2Se} (**Supplementary Fig. S13b** and **c**). For the V_{WSe_x}, we are not able to exactly find the X value. So, we have considered three types of V_{WSe_x} defects such as V_W, V_{WSe3}, and V_{WSe6} (**Supplementary Fig. S13b** and **c**). For the antisite defect, we have observed both W_{Se}, and Se_W (or 2Se_W). Indeed, there would be other cluster defects especially at the grain boundaries. We have rarely observed some other topological defects (**Fig. R4-2**) such as 5 | 7 rings and dislocation chain mostly happening at the grain boundaries as observed in the other studies on ML TMDC materials [*Nano Lett.* 13, 2615-2622 (2013)]. However, our photocatalytic CO₂ reduction results have revealed that the activity

of the edge region is two orders of magnitude larger than the basal plane (as discussed in the Comment#5). So, we have not focused more on the rarely observed cluster defects on the basal plane for the DFT calculations which requires a separate set of detailed investigations.

Fig. R4-2: HRTEM images of the cluster defects. 5 | 7 rings and dislocation chain at the grain boundaries.

Changes: Supplementary Fig. S13 (in the revised Supplementary Information) has been re-plotted.

(3) Figure S14 nicely shows the phase diagram depicting stable defects at different Se chemical potentials. However, the paper is a bit too light on background. How was the formation energy calculated – provide an equation. Chemical potential is a rather ‘unintuitive’ quantity, which is for gases usually converted to temperature and pressure. How about for Se? What do the investigated values of selenium chemical potentials compare to?

Response: We appreciate the reviewer for pointing out a lack of information in our calculations of formation energy. By the definition of the chemical potential, the total Gibbs energy of the ML WSe_2 can be represented as: $G(\text{ML } WSe_2) = n\mu_W + 2n\mu_{Se}$, where n is the number of WSe_2 units, μ_W and μ_{Se} are the chemical potentials for a W and Se atoms, respectively. In our calculations, the temperature and pressure dependence are ignored with acceptable accuracy due to that W and Se elements exist in the solid phase [*Phys. Rev. B Condens. Matter.* 38, 7649-7663 (1988) & *Phys. Rev. Lett.* 67, 2339-2342 (1991)]. Under the approximation, the chemical potential is the calculated total energy per atom at $T = 0$. Hence, the chemical potential of bulk Se is calculated as $\mu_{Se}(\text{bulk}) = \frac{E_{Se}(\text{bulk})}{n_{Se}}$, where $E_{Se}(\text{bulk})$ is the total energy of bulk Se with the trigonal structure. For the ML WSe_2 , the chemical potential of Se has been linked to the chemical potential of W as $\frac{E(\text{ML } WSe_2)}{n} = \mu_W + 2\mu_{Se}$. The investigated values of Se chemical potentials are compared to the chemical potential of the bulk Se and are restricted to the calculated enthalpy of the ML WSe_2 . The calculated range for the change of the Se chemical potential ($\Delta\mu_{Se}$) spans from 0 to -0.63 eV, which corresponds to the formation of the bulk Se ($\Delta\mu_{Se} = 0$ eV) and the formation of the bulk W ($\Delta\mu_{Se} = -0.63$ eV).

Changes: The statements about the detailed background and equations have been added in the revised Supplementary Information (section “**Supplementary Note 4–6. Calculation of formation energy**”). All the modifications have been highlighted (YELLOW color) in the text.

(4) Figure S15, S18: As said in the introduction, I really do not trust DOS at the PBE level for semiconductors. Please re-do at a hybrid level.

Response: We agree with the reviewer’s suggestion. The GGA is well-known for underestimating band gaps of semiconductors, and even in some cases, a vanishing band gap is obtained, which is not suitable to determine the semi-metal and metallic properties for our system. For describing the electronic structure more correctly, we have re-done the DOS by using the HSE06 hybrid functional based on the optimized structure from the PBE level (**Fig. R4-3** and **Fig. R4-4**).

Fig. R4-3: Electronic properties of the basal plane with the intrinsic defects. Density of states (DOS) of the pristine basal plane and different intrinsic defects in the basal plane. Both V_{WSe6} and W_{Se} show a non-zero magnetic moment of 2.00 μ_B per supercell.

Fig. R4-4: Electronic properties of the reconstructed edges. Total local DOS (red line) of the ZZ, An, and AC edges, as shown in Supplementary Fig. S17. The DOSs of the middle regions of the model ribbon are plotted in the background by blue lines.

Changes: The statements about the calculation of DOS at hybrid level have been added in the revised Supplementary Information (2nd paragraph of section “**Supplementary Note 4-1. Computational method**”). **Supplementary Fig. S16** and **Fig. S19** (in the revised Supplementary Information) have been re-plotted based on the distinguished reviewer’s suggestion. All the modifications have been highlighted (YELLOW color) in the text.

(5) There seems to be no charge transfer from the surface to the CO₂ molecule. Values for E_{ads} around -0.4 eV hint at physisorption. This is what the authors correctly identified in the manuscript. Where is the molecule then activated?

Response: We thank the reviewer very much for pointing this out. We agree with the distinguished reviewer that the CO₂ molecule cannot be activated on the basal plane (surface) of the ML WSe₂. However, our previous analyses show a small activity for the basal plane. Therefore, we have performed the blank tests to check if the activity of the basal plane is real or an artifact (**Fig. R4-5**) [*Chem. Eur. J.*, 24, 12739-12746 (2018)]: (i) With WSe₂ photocatalyst/with CO₂/without light (with the production of Y_{b1}), (ii) With WSe₂ photocatalyst/without CO₂/with light (with the production of Y_{b2}), and (iii) Without WSe₂ photocatalyst/with CO₂/with light (with the production of Y_{b3}). Finally, the values of products measured in the blank experiments have subtracted from the product yield in the photocatalytic reaction as,

$$Y = Y_{exp} - Y_{b1} - Y_{b2} - Y_{b3}. \quad (1-1)$$

where Y is the total blank-corrected yield. Regarding these corrections, the total methane yields (**Fig. 4f** in the revised manuscript) and IQE s (**Fig. 4g** in the revised manuscript) have been modified (Notably the highest IQE value decreased from 0.29% to 0.23% after the subtraction of the blank test). Then, we have re-fitted the IQE plot with $f_e = 0.23$ and $f_b = 0.007 f_e$, showing that the activity of the basal plane is in the order of the blank experiments. It is in well-agreement with our DFT calculations and SECM results that reveal the basal plane of the ML TMDCs is inert, which is consistent with the other reports [*Science*, 317, 100-102 (2007) & *Nat. Mater.*, 15, 48-53 (2016) & *Nat. Commun.*, 8, 15113 (2017)]. Therefore, the new fitting of the IQE has resulted in a higher contribution for the edge sites (more than two orders of magnitude compared with the basal plane) in the photocatalytic activity of the ML WSe₂. Moreover, we have re-calculated the consumed electron rate of $3.8 \pm 0.7 e^- s^{-1}$ per edge atom.

Fig. R4-5: Photocatalyst yields and blank tests. Total yield for methane (CH_4) and acetaldehyde (CH_3CHO) products after 4 h under one sun irradiation. Our results show that acetaldehyde is a minor product because its yield is in the order of the blank tests.

Changes: The statements about the blank tests have been added to the **Supplementary Notes 6–1** in revised Supplementary Information. **Supplementary Fig. S25** and reference [*Chem. Eur. J.*, 24, 12739–12746 (2018)] have been added in the revised Supplementary Information. Accordingly, all the reported yields (**Fig. 4f** in the revised manuscript), *IQEs* (**Fig. 4g** in the revised manuscript), the simulated yields of PC CO_2RR (**Supplementary Fig. S28c** and **Supplementary Table S1** in the revised Supplementary Information), and the consumed electron rate per edge atom (**Supplementary Table S2** and **Supplementary Notes 6–3** in the revised Supplementary Information) have been modified all over the revised texts. All the modifications have been highlighted (YELLOW color) in the text.

(6) In the manuscript, it is not really clear what different terminations, edges etc. represent and which geometries appear in the DFT calculations. Please provide a clearer way of showing that, maybe through a table (can be in the SI).

Response: We thank the reviewer very much for pointing this out. We have replotted the configuration of the edge reconstruction (**Fig. R4-6**) and studied regular edge (**Fig. R4-7**) by DFT calculation.

Fig. R4-6: Edge reconstruction by DFT calculation. Top-view and cross-sectional images of the relaxed structures of the edges for different edges: **a** to **c**, Terminated edge with ZZ_W/ZZ_{Se} (ZZ), An_{Se}/An_W (An), and AC, respectively. ZZ, An, and AC stand for zigzag, antenna, and armchair. Blue- and dark orange-filled circles stand for W and Se atoms. Similarly, An_{Se}/ZZ_W and ZZ_{Se}/An_{Se} configurations were calculated as well (not shown here).

Fig. R4-7: Regular edge by DFT calculation. Top-view and cross-sectional images of the relaxed structures of the edges for different edges: Terminated edge with ZZ_W/ZZ_{Se} (Z_1), An_{Se}/An_W (Z_2), An_{Se}/ZZ_W , ZZ_{Se}/An_{Se} , and AC, left to right. ZZ, An, and AC stand for zigzag, antenna, and armchair. Blue- and dark orange-filled circles stand for W and Se atoms. Shadow regions show the fixed atoms in each model.

Moreover, we have provided a new table (**Table #1**) for the configurations of the defective ZZ and An edges.

Table #1: Models of defective edges. ZZ_{Se} and An initial input configurations with different defects are shown and explained.

Structure	Explanation	Figure
ZZ_{Se}	Input file	$ZZ_{Se} + V_{2Se}$	Se (1) and Se (2) are removed	Fig. 2c (Manuscript)
$ZZ_{Se} + W_{Se}$	Se (1) replaced by W (relaxed configuration I)	Fig. 2c (Manuscript)
$ZZ_{Se} + W_{add}$	W is added between Se (1) and Se (3)	Fig. 2c (Manuscript)
$ZZ_{Se} + W_{Se,II}$	Se (1) replaced by W (relaxed configuration II)	Supplementary Fig. S18
$ZZ_{Se} + W_{Se,III}$	Se (4) replaced by W (symmetric relaxed structure)	Supplementary Fig. S18
$ZZ_{Se} + W_{Se,IV}$	Se (4) replaced by W (asymmetric relaxed structure)	Supplementary Fig. S18
An_{Se}	Input file	$An + V_{2Se}$	Se (1) and Se (2) are removed	Fig. 2c (Manuscript)
$An + W_{Se}$	Se (1) replaced by W	Fig. 2c (Manuscript)
$An + W_{add}$	W is added between Se (1) and Se (3) (relaxed configuration I)	Fig. 2c (Manuscript)
$An + W_{Se,II}$	Se (4) replaced by W (relaxed configuration I)	Supplementary Fig. S18
$An + W_{Se,III}$	Se (4) replaced by W (relaxed configuration I)	Supplementary Fig. S18
$An + W_{add,II}$	W is added between Se (1) and Se (3) (relaxed configuration II)	Supplementary Fig. S18

Changes: Supplementary Fig. S25, Supplementary Note Fig. N5, and Supplementary Note Table N1 have been added in the revised Supplementary Information.

Important: The authors should make it clear that they made no attempt to capture the excited states and photocatalysis in their DFT calculations. This is acceptable since DFT is a tool for describing ground states (and is occasionally “misused” to calculate higher states by promoting some electrons) but it should be acknowledged. Limitations of such calculations must be discussed.

Response: Thanks for the reviewer’s suggestion. The purpose of our DFT calculations is to determine the stable geometries of our ML WSe₂ flakes and the DOS at the electronic ground state. Based on those properties, we further determine the active sites for CO₂ adsorption without the interaction with light, which is regarded as the first step to understand CO₂ activation and conversion. All the DFT results here are carried out for the electronic ground state.

Changes: These statements have been added in the section “**Calculation**” (3rd paragraph of the section “**Method**”) of the revised manuscript. All the modifications have been highlighted (YELLOW color) in the text.

If the authors could provide a little bit more information on the calculations, did single point calculations at a hybrid level at least for the electronic properties (adsorption energies and geometries are probably fine at the PBE level) and show the structures explicitly (XYZ files), I would have no problem with the DFT part of the paper. A revision is recommended.

Response: We thank Referee #4 for the suggestions and positive evaluation. We hope that the distinguished reviewer will find our responses as detailed above to his/her comments satisfactory.

REVIEWER COMMENTS

Reviewer #1 (Remarks to the Author):

I think the authors have resolved all the issues and this paper can be published now.

Reviewer #2 (Remarks to the Author):

The authors have addressed all of my comments. The paper has been very much improved. In particular, the large set of blank tests is very much appreciated.

There is one last comment on the oxidation half reaction that remains to be clarified:

(1) Can the amount of oxygen be quantified, or at least an order-of-magnitude estimation be performed? The yield of acetaldehyde in the blank tests is also surprising. Is it possible that acetaldehyde is formed from a photocatalytic oxidation reaction of an impurity, which would also consume holes? Can the authors comment?

After this last minor revision has been clarified, the paper can be accepted without another round of peer-review.

Reviewer #3 (Remarks to the Author):

I think that the paper is not fully convincing and thus ready for publication

Reviewer #4 (Remarks to the Author):

The authors did a satisfactory job of addressing my (and other reviewers') previous comments.

I suggest that the paper be published.

However, an inconsistency needs to be rectified.

1) The authors have redone some of DFT at the hybrid level, as suggested. However, this information is buried in the SI, while in the manuscript (Lines 305-319) this information is not presented. Discussing only the PBE level of DFT in the manuscript gives a misleading impression that the DOS was also done at this level, while it clearly wasn't (evident from the SI). Please update.

REVIEWER COMMENTS

Reviewer #1 (Remarks to the Author):

I think the authors have resolved all the issues and this paper can be published now.

Reviewer #2 (Remarks to the Author):

The authors have addressed all of my comments. The paper has been very much improved. In particular, the large set of blank tests is very much appreciated.

There is one last comment on the oxidation half reaction that remains to be clarified:

(1) Can the amount of oxygen be quantified, or at least an order-of-magnitude estimation be performed? The yield of acetaldehyde in the blank tests is also surprising. Is it possible that acetaldehyde is formed from a photocatalytic oxidation reaction of an impurity, which would also consume holes? Can the authors comment?

After this last minor revision has been clarified, the paper can be accepted without another round of peer-review.

Reviewer #3 (Remarks to the Author):

I think that the paper is not fully convincing and thus ready for publication

Reviewer #4 (Remarks to the Author):

The authors did a satisfactory job of addressing my (and other reviewers') previous comments.

I suggest that the paper be published.

However, an inconsistency needs to be rectified.

1) The authors have redone some of DFT at the hybrid level, as suggested. However, this information is buried in the SI, while in the manuscript (Lines 305-319) this information is not presented. Discussing only the PBE level of DFT in the manuscript gives a misleading impression that the DOS was also done at this level, while it clearly wasn't (evident from the SI). Please update.

RESPONSES TO THE REVIEWERS' COMMENTS

We thank the reviewers for taking the time to review our manuscript comprehensively. We have addressed their comments, which we find very relevant and important, and we have modified the paper accordingly.

We provide a point-by-point response (in **black** color) to the reviewers' comments and remarks (in **blue** color) below and explain the corresponding changes and additions that we have made to properly address them in the revised version of the manuscript and Supplementary Information.

REVIEWERS' COMMENTS

Reviewer #2 (Remarks to the Author):

The last issue is now also resolved and the paper can be accepted for publication.

REVIEWERS' COMMENTS

Reviewer #2 (Remarks to the Author):

The last issue is now also resolved and the paper can be accepted for publication.

Referee #2

The last issue is now also resolved and the paper can be accepted for publication.

Response: We appreciate the distinguished reviewer for the very positive recommendation of our work.